# Attenuated Total Reflection at THz Wavelengths: Prospective Use of Total Internal Reflection and Polariscopy

Meguya Ryu [1,2,*], Soon Hock Ng [3], Vijayakumar Anand [3], Stefan Lundgaard [3], Jingwen Hu [3], Tomas Katkus [3], Dominique Appadoo [4], Zoltan Vilagosh [5], Andrew W. Wood [5], Saulius Juodkazis [3,6,*] and Junko Morikawa [2,*]

1   Research Institute for Material and Chemical Measurement, National Metrology Institute of Japan (AIST), Tsukuba Central 3, 1-1-1 Umezono, Tsukuba 305-8563, Japan
2   CREST-JST and School of Materials and Chemical Technology, Tokyo Institute of Technology, 2-12-1 Ookayama, Meguro-ku, Tokyo 152-8550, Japan
3   Optical Sciences Centre and ARC Training Centre in Surface Engineering for Advanced Materials (SEAM), School of Science, Swinburne University of Technology, Hawthorn, VIC 3122, Australia; soonhockng@swin.edu.au (S.H.N.); vanand@swin.edu.au (V.A.); slundgaard@swin.edu.au (S.L.); jhu@swin.edu.au (J.H.); tkatkus@swin.edu.au (T.K.)
4   THz Beamline, ANSTO-Australian Synchrotron, Clayton, VIC 3168, Australia; Dominique.APPADOO@ansto.gov.au
5   Australian Centre for Electromagnetic Bioeffects Research, Swinburne University of Technology, Hawthorn, VIC 3122, Australia; zvilagosh@swin.edu.au (Z.V.); awood@swin.edu.au (A.W.W.)
6   World Research Hub Initiative (WRHI), School of Materials and Chemical Technology, Tokyo Institute of Technology, 2-12-1 Ookayama, Meguro-ku, Tokyo 152-8550, Japan
*   Correspondence: ryu.meguya@aist.go.jp (M.R.); sjuodkazis@swin.edu.au (S.J.); morikawa.j.aa@m.titech.ac.jp (J.M.)

**Abstract:** Capabilities of the attenuated total reflection (ATR) at THz wavelengths for increased sub-surface depth characterisation of (bio-)materials are presented. The penetration depth of a THz evanescent wave in biological samples is dependent on the wavelength and temperature and can reach 0.1–0.5 mm depth, due to the strong refractive index change ∼0.4 of the ice-water transition; this is quite significant and important when studying biological samples. Technical challenges are discussed when using ATR for uneven, heterogeneous, high refractive index samples with the possibility of frustrated total internal reflection (a breakdown of the ATR reflection mode into transmission mode). Local field enhancements at the interface are discussed with numerical/analytical examples. Maxwell's scaling is used to model the behaviour of absorber–scatterer inside the materials at the interface with the ATR prism for realistic complex refractive indices of bio-materials. The modality of ATR with a polarisation analysis is proposed, and its principle is illustrated, opening an invitation for its experimental validation. The sensitivity of the polarised ATR mode to the refractive index between the sample and ATR prism is numerically modelled and experimentally verified for background (air) spectra. The design principles of polarisation active optical elements and spectral filters are outlined. The results and proposed concepts are based on experimental conditions at the THz beamline of the Australian Synchrotron.

**Keywords:** ATR; THz; synchrotron radiation; diagnostics; polariscopy; four polarisation method

## 1. Introduction

Transmission spectroscopy of biological samples at the long-IR and THz wavelength have to be made in a hydrated state [1]. Water absorption can be significant, especially in the THz spectral range and when high brilliance synchrotron radiation is utilised [2]. There is a particular advantage to using the attenuated total reflection (ATR) technique at THz wavelengths, where water absorption can be detrimental to free space THz beam propagation. With minimal sample preparation, its placement onto an ATR diamond window (in this study) offers a simple and effective method of material characterisation.

Harnessing the THz portion of synchrotron radiation, high brightness THz beams are extended to <300 cm$^{-1}$ wavenumbers (single digit THz frequencies), where conventional tabletop Fourier transform IR (FTIR) spectrometers are rendered less useful, due to the low signal-to-noise ratio (SNR). The high brilliance of THz synchrotron radiation makes it especially promising for real-time monitoring of phase transitions. Due to the high transparency of diamond at THz frequencies, the characterisation of water at high pressure was demonstrated in [3]. An anvil and pressure cell (using diamond windows) experiment with materials at high pressure and temperature was recently demonstrated with a 35 MPa cell at sub-THz spectral window [4]. Charge separation and its control over the photosynthetic reaction centres upon illumination and is another active area of research where THz spectroscopy of bio-molecules and proteins can provide new insights [5]. The THz properties of protein structures and their hydration shells were studied extensively [6–8]. The complexity increases when coupled with the temperature-dependent properties of water [9]. When the ATR technique is used with a high brilliance THz-IR beam (Australian Synchrotron), it becomes possible to measure the THz response of both simple and complex composition samples and to reveal temporal changes. High quality scans can be produced every 0.3–0.4 s, allowing for the monitoring of effects caused by rapid temperature changes. The typical signal-to-noise ratio (S/N) ratio is ~20 times higher as compared with Globar sources widely used in table-top FTIR spectrometers.

The water–ice transition is of particular interest for cryogenic applications in biological materials. It can also be studied and exploited using the ATR technique for deeper characterisation of bio-samples. Due to both a large refractive index change from the water–ice transition (~0.4) and change in absorption coefficient through the THz range, the ATR condition can be not fulfilled, i.e., instead of reflection from the diamond-sample interface, the THz beam propagates in transmission. The total internal reflection occurs for the light propagating from the medium with a higher refractive index into the lower index material (ATR diamond to sample). A higher refractive index of the sample (with ice) invalidates the ATR performance. The refractive index of ice is ~1.7 at 1-to-3 THz, increases to ~1.9 at 4.2 THz, reduces slowly to 1.75 at 6.5 THz, then quickly drops to 1.1 by 6.9 THz [10]. This means that between 1 and 3 THz, there is a good refractive index contrast between liquid water and ice. Due to the air gaps between the sample and the ATR diamond window, light tunnelling can take place, also contributing to the transmission mode. Addressing these issues and taking advantage of (some of) them is discussed in this prospective/concept paper with numerical examples and required analytical expressions.

In this perspective/concept paper, based on recent experiments at the THz beamline at the Australian Synchrotron, technical peculiarities of the ATR operation mode are detailed. We introduce the concept of polarisation analysis for the ATR mode and validate it numerically and experimentally. Challenges of multi-reflection beam delivery to (and from) the sample in ATR measurements, as well as ATR condition changes with the liquid–solid water transition, are scrutinised in order to extract the useful orientation characterisation of samples in ATR experiments. Polarisation, intensity and the spectral filtering of complex synchrotron THz radiation can provide more flexibility in material characterisation.

## 2. Peculiarities of ATR Measurements

The ATR condition can be altered by a refractive index change in the sample due to the phase transition of ice to water, or due to light tunnelling through small air gaps between the surface of the ATR prism and the sample. Even then, the ATR setup can be made useful by applying a top reflecting mirror. These conditions are discussed below together with spatial intensity-spectral filters, which can help to streamline THz band detection in the sample at specific wavelengths. Finally, the orientation of a sample can be exploited together with polarisation in the ATR setup, which is currently not used to its full potential. Recent results based on measurements using synchrotron-ATR at the Australian Synchrotron are presented, and the concept of polarisation-discriminated measurements is outlined.

### 2.1. Attenuated Total Reflection

The technique of attenuated total reflection (ATR) spectroscopy has become a popular method to measure the dielectric properties of materials [11,12]. It relies on total internal reflection within a prism and the resultant evanescent wave that extends beyond the ATR prism surface. A sample is placed on this surface, and the incident energy is absorbed only via the evanescent wave generated between the sample and ATR prism. As such, there is a reduction in the reflected signal intensity at the total reflection angle, when there is no transmitted light into the sample (placed on top of the ATR prism). The advantage of the method lies in the fact that solid and liquid samples can be studied with minimal preparation. This is exceptionally useful for the THz spectral window since free space propagation of THz radiation under normal conditions is strongly attenuated, due to absorption by water vapour. The observed ATR spectrum is noted to be nearly equivalent to that of the transmission mode. If the absorption of the evanescent wave is negligible, the penetration depth of the evanescent E-field ($1/e$-level) is given by the following [13]:

$$d_p^{(ev)} = \frac{\lambda}{2\pi n_1 \sqrt{\sin^2 \theta_i - (\frac{n_2}{n_1})^2}},\qquad(1)$$

where $\lambda$ is the wavelength, $n_1$ and $n_2$ are the real parts of the refractive index of the ATR crystal and sample, respectively, and $\theta$ is the angle of incidence of the incoming radiation; for the beam intensity $I = E^2$, the penetration depth becomes $d_p^{(ev)}/2$. Since $\sqrt{\sin^2 \theta_i - (n_2/n_1)^2}$ has to be a real number, $n_2 < n_1 \sin \theta_i$ sets the limit on the refractive index of the sample for a given ATR crystal. ATR finds its use at infrared frequencies of 30 to 400 THz, where the $d_p$ is of the order of 1 $\mu$m for biological tissue. At these distances, the total absorption of the evanescent wave at distances in the order of the $d_p$ is, indeed, negligible. However, at low terahertz frequencies (0.3 to 3 THz) the wavelength $\lambda$ is on the order of 1.0–0.1 mm, which yields a $d_p$ on the order of 0.02 to 0.2 mm. The absorption of the evanescent wave becomes not negligible at such depths in biological tissues. A more elaborate equation for the reflected wave intensity, using the complex refractive index $n^* = n + ik$, where $k$ is the imaginary part of the complex refractive index $n^*$, is required when polarisation becomes important.

It is instructive to estimate what are frustrated—the total internal reflection (TIR) transmission predictions. This affects the intensity reaching different depths of the sample as well as accounting for the detected THz power losses when an air gap is formed between the sample and the ATR prism. The fraction of light intensity polarized in the plane of incidence (p-pol) at the wavelength $\lambda$, transmitted across the gap of width $d$, with refractive index $n_{gap} = 1$, and in the case of the ATR prism of refractive index $n$, is given as follows [14] (Figure 1):

$$T_p = \frac{1}{\alpha \sinh^2 y + 1},\quad \alpha = \left(\frac{n^2 - 1}{2n}\right)^2 \frac{[(n^2 + 1)\sin^2 \theta_i - 1]^2}{\cos^2 \theta_i (n^2 \sin^2 \theta_i - 1)},\quad y = 2\pi\left(\frac{d}{\lambda}\right)\sqrt{n^2 \sin^2 \theta_i - 1},\qquad(2)$$

where $\theta_i$ is the angle of incidence onto the first surface (ATR prism); the same material of refractive index $n$ is considered on both sides of the gap. This set of formulas also shows that a larger light penetration depth can be achieved if a sample on the ATR prism has high refractive index inclusions. When the inclusion has a larger refractive index, compared to its surroundings, the frustrated TIR conditions applies, and light tunnels through the separation with a lower refractive index. This situation is illustrated numerically in Figure 1b. Larger sub-wavelength depths in the sample can be optically excited by light tunnelling in the sub-millimetre wavelength region.

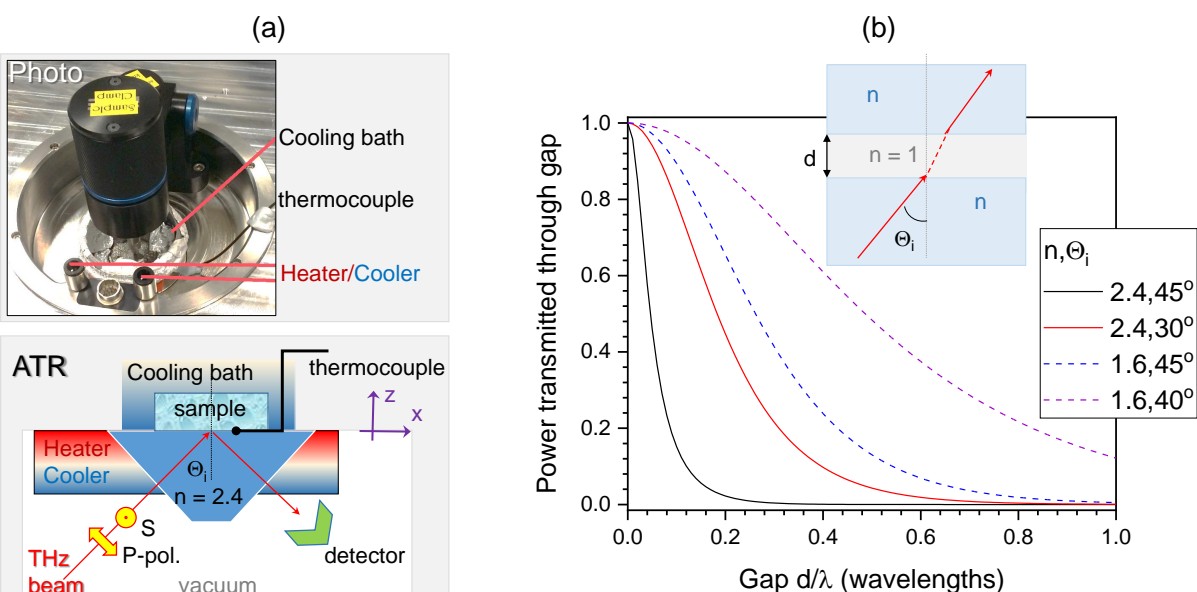

**Figure 1. (a)** Photo and schematics of ATR setup used for temperature dependent absorbance spectroscopy at the THz/Far-IR beamline [15,16]. **(b)** Visualisation of frustrated TIR (Equation (2)) for different angles of incidence and refractive indices $n$ of the slabs. The index $n = 1.6$ also corresponds to the case when two $n = 2.4$ (diamond) slabs are separated by $n_{gap} = 1.5$ (bio-sample), i.e., $\frac{n}{n_{gap}} = \frac{2.4}{1.5} = 1.6$.

We demonstrate the ATR technique for determination of the THz penetration depth into bio-materials at different temperatures, using synchrotron radiation (the Australian Synchrotron) in the frustrated reflection mode with the Au-mirror placed above the 2 mm-thick sample [15,16].

### 2.2. A Useful Condition When Atr Becomes Invalid

When measuring the reflection of THz radiation in the ATR geometry at $\theta_i = 45°$, the ratio of the refractive indices of the diamond prism $n_1 = 2.42$ and sample $n_2$ is important since the ATR occurs at $n_1 \sin \theta_i > n_2$ (Equation (4)). This sets $n_2 < 1.71$ and is critically important in the characterisation of bio-materials close to this ATR condition (a heating/cooling compatible ATR unit is shown in Figure 2a,b). With water $n_2 \approx 2.1$ and ice $n_2 \approx 1.7$ at $\approx 1$ THz, direct transmission into the sample takes place (Figure 2c). This change in the refractive index can be exploited for better depth characterisation of materials, due to the larger penetration depth of the evanescent field and direct transmission into the sample. The latter case requires modification of the measurement technique via the addition of a top reflecting mirror to redirect THz radiation back to the detection direction along the ATR geometry (Figure 2c) as we introduced recently [16].

The optical path length $l_T$ in the case of the transmission mode in the ATR setup is defined by the distance, $H$, to the mirror (see drawing in Figure 2c) as $l_T = 2H/\cos(\theta_2)$ (from Snell's law $n_1 \sin(\theta_1) = n_2 \sin(\theta_2)$ and the Pythagorean identity of sin and cos functions, one finds $\cos \theta_2 = \sqrt{1 - \left(\frac{n_1}{n_2} \sin \theta_1\right)^2}$). Precise determination of $H$ is important since the surfaces of the ATR prism and sample housing unit usually have height differences of tens-of-micrometers; Figure 2a,b. Power reflected at the first prism–sample interface is $I(R_1) = RI_{in}$, where $R$ is the energy reflection coefficient and $I_{in}$ is the incident power. Considering a perfect reflection $R_{Au} = 1$ at the top-mirror, the power re-directed back to the detector is $I(R_2) = (1 - R)^2 I_{in} \exp(-\alpha l_T)$.

The experimentally measured ratio $X_{exp} \equiv \frac{Power_{Sample/Diamond}}{Power_{Air/Diamond}} = (I(R_1) + I(R_2))/I_{in} = R + (1 - R)^2 \exp(-\alpha l_T)$. The absorption (optical losses) coefficient is then $\alpha = -\frac{\cos(\theta_2)}{2H} \ln\left(\frac{X_{exp} - R}{(1 - R)^2}\right)$. The reflectivity coefficient (power, intensity) is defined by the angle of inci-

dence $\theta_i$ and the complex refractive indices of sample and ATR-prism $\tilde{n} = n + i\kappa = \sqrt{\varepsilon}$, where $\varepsilon$ is the permittivity.

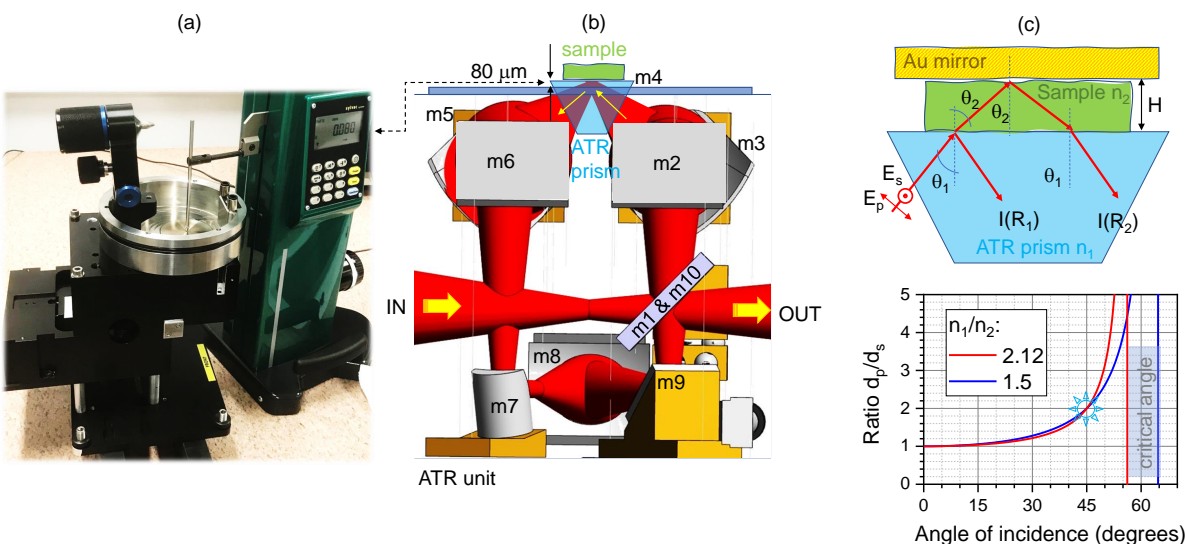

**Figure 2.** (**a**,**b**) Photo and schematics of the beam path in ATR unit. Three reflections are encountered from the IN-port to the sample (mirrors m1–3) and extra six for the path sample-to-OUT port (m5–10). (Image: courtesy Dr. Jeff Kuehl) (**c**) Schematics of measurements with top reflecting mirror when TIR conditions are not fulfilled due to the high refractive index $n_2$ of the sample. Plot shows the evanescent field ratio for the p-pol. (TM) and s-pol. (TE) $d_p/d_s$ (Equation (3)) vs. the angle of incidence $\theta_i$. For the non-polarised illumination, the penetration depth of evanescent field is $d_{ev} = \frac{d_s + d_p}{2}$. Critical angle is at $\theta_c = \arcsin\left(\frac{n_2}{n_1}\right)$.

For the normal incidence and lossless ATR diamond prism ($n_1, \kappa_1 = 0$), the reflectivity from the sample ($n_2, \kappa_2$) is $R = [(n_1 - n_2)^2 + \kappa_2^2]/[(n_1 + n_2)^2 + \kappa_2^2]$ and the absorbed part of energy is $A = 1 - R$ since there is no transmission $T = 0$. The Beer–Lambert law defines the transmitted part $T = e^{-\alpha l_T}$ in a single pass through thickness $l_T$; $\alpha l_T \equiv \ln(10)OD$, where the optical density (absorbance) is calculated from transmittance $OD = -\lg T$ (note that this expression can be used for back-reflected signals in the ATR setup with a top mirror when the transmission is collected in the reflection direction and the ATR condition is not satisfied). For an arbitrary angle of incidence $\theta_i$, the reflectivity for s-/p-pol. can be expressed as $R_s = r_s^2 = \left(-\frac{\sin(\theta_1 - \theta_2)}{\sin(\theta_1 + \theta_2)}\right)^2$ and $R_p = r_p^2 = \left(\frac{\tan(\theta_1 - \theta_2)}{\tan(\theta_1 + \theta_2)}\right)^2$, Fresnel's sine and tangent laws, respectively.

The described measurement with the invalid ATR condition using a top-mirror allows deeper material interrogation, owing to the absorption from a much longer optical path as compared to that probed by the evanescent field (at ATR conditions) as shown in the plot in Figure 2c. The penetration depth into the sample at ATR conditions is the sub-wavelength and the s-pol. probes half the depth as compared to p-pol. at the specific $\theta_i = \pi/4$ incidence, regardless of the relative refractive index $\frac{n_1}{n_2}$. The penetration depth for s- and p-polarisations is as follows [13]:

$$d_s = \frac{\lambda_0}{n_1} \frac{n_{21} \cos\theta_i}{\pi(1 - n_{21}^2)\sqrt{\sin\theta_i^2 - n_{21}^2}}, d_p = \frac{\lambda_0}{n_1} \frac{n_{21} \cos\theta_i[2\sin^2\theta_i - n_{21}^2]}{\pi(1 - n_{21}^2)[(1 + n_{21}^2)\sin^2\theta_i - n_{21}^2]\sqrt{\sin^2\theta_i - n_{21}^2}}, \tag{3}$$

where $n_{21} = n_2/n_1$ (sample/prism), $\lambda_0$ vacuum wavelength, $\theta_i$ is the angle of incidence onto the prism–sample interface.

### 2.2.1. Example: Water in Butter around Water Freezing Conditions

The THz–ATR technique with a top-reflecting gold mirror is used to measure the absorbance of bio-materials [15]. At the chosen ∼0.7–2.2 THz synchrotron radiation band,

polarisation is mostly linear and corresponds to the s-pol. at the sample–prism interface. Freezing water reduces its real part of the refractive index only slightly ($n$ is proportional to mass density $\rho$); however, a very strong reduction in absorption occurs. Experimental values of ice reflectivity [17–19] show a significant increase at the 56.5–75.0 cm$^{-1}$ window (1.70–2.25 THz). Up to three times difference between the reflectivity of water ($\sim$4%) and ice ($R > 12\%$) to that of air is observed [17–19].

Butter (Lurpak, a supermarket brand) has 14.7% $w/w$ water, 3% $w/w$ non-fatty solids, which are dissolved in the water, and the rest is fat [20]. The temperature is varied from −20 to 24 °C over a total time frame of 24 min with continual scanning; a total of 140 sets of 40 averaged scans are collected during that time span. Figure 3 shows the reflectance evolution with temperature at two spectral regions. The inset presents reflectivity changes with temperature. There is a featureless temperature dependent reduction variation with butter, which is evident at the most sensitive frequency range of 0.95 to 1.0 THz. A considerably stronger temperature dependence of reflectivity is observed at the 2 THz band. Overall, the reflected ATR signal for Lurpak butter shows a 13% decrease over the −20 °C to +24 °C change at 1.0 THz and a 3.5% decrease at 2 THz. A slight decrease in reflectance with temperature variation ±20°C around water freezing indicates that the water content ∼14.7% needs to be regarded as being "bound" in a homogenous mixture of fat and protein. The temperature-related reduction in reflectance in the region of 2.0 THz shows a plateau in the 2–12 °C range. These results suggest that other materials with bound water may not show the effect of water freezing.

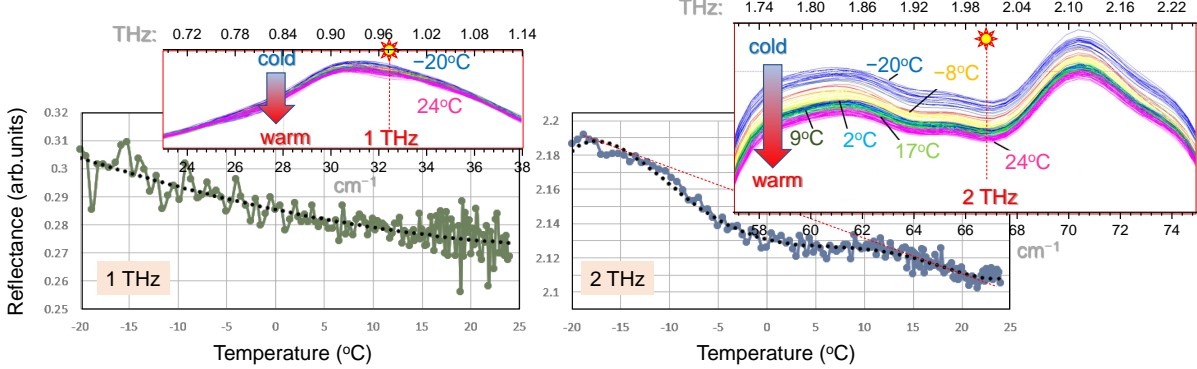

**Figure 3.** Temperature dependence of butter sample (2-mm-thick) at 1 and 2 THz measured with Au top-mirror [15]; wavelength of 1 THz is 0.3 mm and 2 THz is 0.15 mm. The insets show the reflectance intensity vs. the wavenumber; air (no Au top-mirror) reflectance is taken as the reference. Dotted-lines (black) are eye guide splines of the experimental data.

### 2.3. Polarisation and Field Enhancement at the Interface of ATR Prism

A unique feature of ATR geometry is that the maximum intensity of the incident beam, regardless of its polarisation, is localised at the interface (see details below and Figure 4). This is defined by the conditions when light propagates from high-to-low refractive index (even at low angles of incidence when the ATR condition is not fulfilled). For normal beam propagation from low-to-high index (incidence from air), the intensity maximum is $\lambda/4$ above the surface due to interference between the incident and reflected beams. Maximum intensity at the interface using the back-side illumination is used in Raman scattering sensing and light-induced back-side wet etching (LIBWE) in subtractive machining [21,22].

The polarisation of the THz beam is not controlled, as the beam is delivered to the ATR prism. This is the condition when maximum intensity is obtained on the sample. The experimentally determined polarisation of synchrotron radiation at THz/far-IR beamline of the Australian Synchrotron (see Appendix B) is 90% linear and 10% circular, due to the sum of the linear and circular polarisations at <100 cm$^{-1}$. This translates to the prevailing s-pol. on the sample placed on the ATR window for the measurements. For wavenumbers larger than 100 cm$^{-1}$, the ratio linear-to-isotropic polarisation is 20–80%.

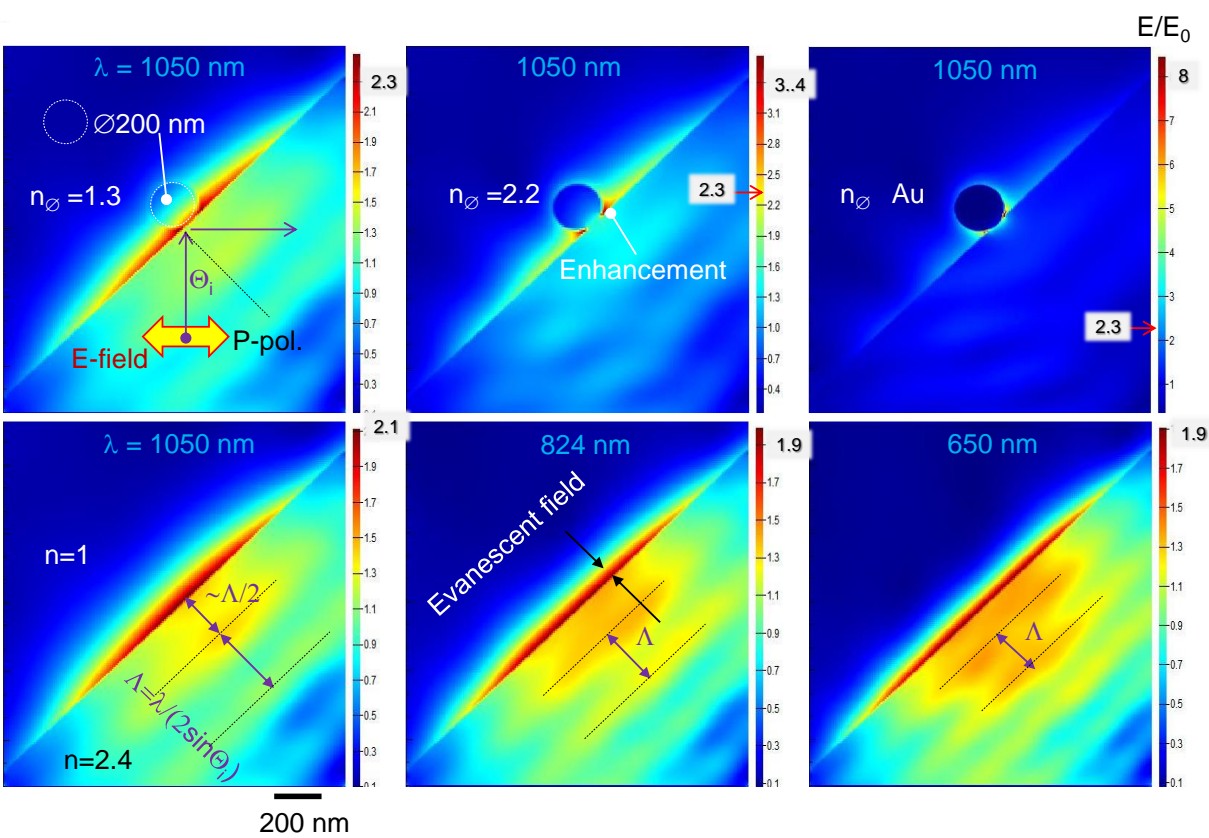

**Figure 4.** Maxwell's scaling (change nm to μm for the same refractive index): numerical finite difference time domain (FDTD; lumerical solutions) calculations for optical wavelengths of $\lambda = 1\,\mu m - 0.65\,\mu m$ to simulate evanescent field at the ATR geometry used (at THz radiation) and at the angle of incidence of $\theta_i = 45°$ (ATR interface is $45°$-tilted). Top row shows E-field enhancement when a $\sim\lambda/20$-diameter sphere of refractive index $n = 1.3$ (water or ice at visible), 2.2 (water or ice at THz) and gold (Au) touches the ATR prism of $n = 2.4$ (diamond); all intensity maps are calculated for the same wavelength $\lambda = 1050$ nm. Bottom row shows the evanescent field at the interface of $\lambda = 1\,\mu m - 0.65\,\mu m$; the interference maxima are recognisable in the $n = 2.4$ media (in the wavelength, there is $\lambda/n$). Calculations are carried out for the p-pol. E-field with strength $E_0 = 1$; hence, $E/E_0$ scale represents enhancement.

For ATR measurements, the depth of the evanescent field is larger for the p-polarisation as compared with s-pol (Figure 2c; Equation (3)). It is important in the case of the frustrated total internal reflection (TIR) modelled for the non-polarised light by Equation (2) (see Figure 1b). In addition, E-field enhancement for the component of the E-field perpendicular to the interface between two dielectrics can be harnessed. Indeed, the boundary conditions of the Maxwell equations require that the normal component of the displacement $D_n = \varepsilon E_n$ be continuous across the boundary (when there are no surface charges), and $\varepsilon$ is the permittivity at the corresponding wavelength—the $E_n = E_{p-pol.}\cos\theta_i$. Hence, the $D_n$ in both the air and diamond are the same, which defines the E-field enhancement $E_{air} = \frac{\varepsilon_{diamond}}{\varepsilon_{air}} \times E_{diamond} \approx 2.4^2 E_{diamond}$ since $n = \sqrt{\varepsilon}$. Note that here, we assume the E-field inside the ATR prism as a reference to calculate enhancement at the air–diamond interface (on the air side). For comparison, reflection at the normal incidence from the vacuum onto the diamond prism with $n = 2.4$ has $R = \frac{(n-1)^2}{(n+1)^2} \approx 17\%$ defining the transmitted $T = 1 - R$ fraction of THz radiation (would be inside the ATR prism).

The penetration depth of the evanescent field depends on the wavelength of light in the ATR geometry (see the bottom row in Figure 4). The intensity distribution in the case of frustrated TIR depends on the normalised gap $d/\lambda$ (Equation (2); Figure 1b). It shows that depending on the angle of incidence and refractive indices, light transmission can take place only when gaps are sub-wavelength. In standard ATR measurements, such frustrated TIR conditions can take place due to the presence of structures and different inclusions of

a higher refractive index in the sample. The wavelength scaling in frustrated TIR gives insights into light tunnelling, regardless of the wavelength. Based on this Maxwell's scaling, it is useful to model light transmission, reflection and enhancement at the interface of the ATR prism, using finite difference time domain (FDTD) calculations (lumerical solutions). Figure 4 shows the results for the IR-visible spectra range; however, the predictions are universal as long as the refractive index is the same. To model a sub-wavelength inclusion inside a sample, a sub-wavelength diameter sphere of different refractive indices, dielectric (1.3 and 2.2), as well as metallic (Au) are chosen. Without the sphere, localisation of the evanescent field at the interface protruding into air gap for length comparable with $\lambda/4$ is clearly recognisable. An extra enhancement of the local E-field at the interface occurs in the nano-gap regions around the sphere. This is an illustration of the E-field enhancement of the normal $E_n$ component discussed above. For the ATR measurements, the local field enhancement around inclusions of difference refractive indices would cause a stronger contribution from those regions where the light intensity was larger. Overall, the volume where the light field intensity is augmented around a sub-wavelength inclusion increases (see the same intensity $E = 2.3$ markers in the top-row maps in Figure 1a).

### 2.4. Polarisation Analysis with Atr

Polarisation analysis in ATR measurements can be explored by using in-plane anisotropy of the sample and along the direction of the evanescent field. All 3D orientations can be sampled for changes in the refractive index and absorption. The real and imaginary parts of the refractive index play their part in defining the polarisation at the output of ATR and are discussed in this section.

The popular four-polarisation (4-Pol) method [23] allows determination of the orientation of the absorbing molecular dipoles in the IR fingerprinting spectral range. It can be used to observe a complex molecular arrangement using microscopy and, when combined with high brightness synchrotron radiation, opens the possibility to observe the change in molecular ordering during phase transitions and crystallisation in real time. Monitoring phase transitions at the spectral window for vibrational–rotation energies at <33 cm$^{-1}$ (<1 THz) can provide insights into nanoscale mechanics and material re-organisation, and are not usually accessible at low-brilliance (non-synchrotron) sources. For example, Debye rotation, network stretching, and rocking and wagging librations of water molecules all are active at the low-THz spectral band [24]. Recently, the 4-Pol method was implemented to reveal molecular ordering in paracetamol (type-*II*) micro-crystals, which are better soluble in water [25]. The principle of measuring absorbance at four sample orientations (or beam polarisation) using linearly polarised light is applicable to any spectral range. Patterns of sub-wavelength, $\lambda/50$, features can be revealed by measuring angular dependence of absorbance/transmittance [26]. The separation of transmittance changes due to birefringence, $\Delta n$, and dichroism, $\Delta\alpha$, in polariscopy (with an aligned polariser and analyser) can be realised, due to the difference in angular dependence of those two contributions, using the 4-Pol technique [27].

Our conjecture here is that the 4-Pol method of orientational probing of absorbance could be also adopted for ATR with one light E-field oriented in the plane of the prism surface (s-pol.) and another perpendicular to it (the evanescent field of incident p-pol.). Developing the 4-Pol method for ATR is our next motivation for experiments on the THz beamline at the Australian Synchrotron. The conjecture is based on basic principles of the polarisation analysis and is applicable to any wavelengths in the ATR measurements.

To select the required orientation of linear polarisation, a polarised mesh-grid is used. Polarisation is defined in the plane of incidence as $E_s$ and $E_p$, which are $\perp$ and $\parallel$ to the plane (or TE and TM modes), respectively. By rotating the mesh-grip polariser by an angle, $\varphi$, the linearly polarised power (energy, intensity), $I$, in the plane of incidence, is defined by $I \propto E_s^2 \cos^2 \varphi + E_p^2 \sin^2 \varphi$ components; $\varphi = 0°$ corresponds to the pure $E_s$ light [28]. Depending on the angle $\varphi$, a different ratio of the $E_z$ and $E_y$ near-field (Figure 5) are

created in the sample. Those two components are absorbed differently, depending on the anisotropy of the sample, i.e., alignment of absorbers.

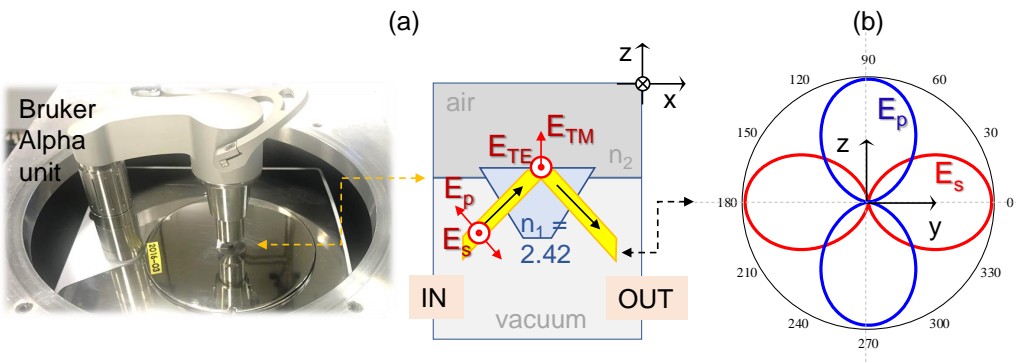

**Figure 5.** (**a**) Geometry and conventions of ATR; photo of the Bruker Alpha unit setup on the THz-beamline compartment. ATR conventions: incident beam (IN) has $E_{s,p}$ polarisations selected by a wire grid polariser incident on the ATR diamond ($n_1$ = 2.42 [29]) prism; plane of incidence is xz. The reflected beam is elliptically polarised due to phase and amplitude changes incurred upon reflection (e.g., air case $n_2 = 1$). The sample is contributing with absorption and phase change for the output beam (OUT). At the interface (sample) the evanescent field $E_{TM}$ (or $E_z$) and $E_{TE}$ or ($E_y$) defines the light–matter interaction. (**b**) The reflected beam can be expressed via $E_{s,p}$ projections measured analyser and given by Equation (4). The angle $\varphi$ is the orientation of the wire grid polariser in the incidence side (IN); $\varphi = 0°$ corresponds to pure $E_s$ component incident onto the ATR prism. The amplitude span in the polar plot is [0–1]. The two components of the $E_p$ and $E_s$ are shown separately by the dipole-like figures (each term in Equation (4)).

The phase and amplitude changes upon ATR reflection (Appendix A) show that one would expect an elliptically polarised light emerging from the ATR prism (even without sample, see inset in Figure A1b and Figure 5). The beam after the prism can be analysed with a linear analyser, which is set to determine $E_s$ and $E_p$ polarisations. The most general expression of the THz power measured after the ATR prism would be the addition of two perpendicular contributions along the s- and p-polarisations:

$$Power^{(OUT)}(\varphi) = A_s \times E_s^2 \cos^2(\varphi + \Delta\psi_s) + A_p \times E_p^2 \sin^2(\varphi + \Delta\psi_p), \tag{4}$$

where $A_{s,p}$ are the arbitrary amplitudes accounting for both absorption (losses) and changes in amplitude due to reflection; $\Delta\psi_{s,p}$ are the phases for s- and p-polarisations, respectively. The angle $\varphi$ defines the orientation of the linear polarisation in the incident beam, i.e., for $\varphi = 0°$, only $E_s$ polarisation is present, and $\varphi = 90°$ corresponds to pure $E_p$. When pure $E_s$ polarisation is incident on the sample and there are no changes in the phase of the reflected beam, the orientational output power will have a horizontally aligned figure-8 (Figure 5) and, consequently, for the pure $E_p$ case, a vertical figure-8. When the incident beam has both $E_s$ and $E_p$ fields, which are absorbed along the z-(evanescent) and y-directions, the emerging field can be modelled with the Equation (4), where changes of phases are also set as fitting parameters. The phase change upon reflection of s-/p-polarisations (TE/TM modes) are solely defined by the real part of the refractive index.

The THz beamline receives ∼20% of a linear $E_L$ (along the slit of the first mirror) polarisation, contributing to $E_s$, and the remaining 80% is isotropic $E_I$ (circular), contributing to the $E_s$ and $E_p$ at frequencies larger than ∼3 THz [28] (see Appendix B for experimental details of the synchrotron beamline). The transmitted power through a metallic linear-grid analyser at the entrance port *IN* to ATR prism is $Power^{(IN)}(\varphi) = \frac{1}{2}(E_I^2 + E_L^2) + \frac{1}{2}E_L^2 \cos(2\varphi) = \frac{1}{2}(0.2 + 0.8) + \frac{1}{2}0.2\cos(2\varphi) \equiv 0.5 + 0.1\cos(2\varphi)$ (shown in Figure 6a). This is an expected angular distribution at the *IN* port if there is no polariser. After the ATR, with no absorption for s- and p-components ($A_{s,p}$) and no phase change

($\Delta\psi_{s,p}$) upon reflection (see Equation (4)), the same angular dependence is expected. However, even without any sample, the beam reflected from the ATR prism will change its polarisation, due to phase changes for s-/p-polarisations $\Delta\phi_{s,p}$, respectively (Equation (4)); see Figure A1b where the s-pol. (TE) experiences an advance of phase by $\sim\lambda/5$, while the p-pol. (TM) reduces the lag of the $2\pi$ phase by a similar amount at the $\theta_i = \pi/4$ angle of incidence on the ATR prism. As a result, just by reflection without any sample, there is a change in polarisation at the OUT port. With the sample, the refractive index contrast (real parts) changes those phase shifts. However, any anisotropy in absorbance $A_{s,p}$ will change the amplitude of the reflected signal at the corresponding polarisation. This would also cause a change in the polarisation of the outgoing beam at OUT port. Equation (4) accounts for all those changes.

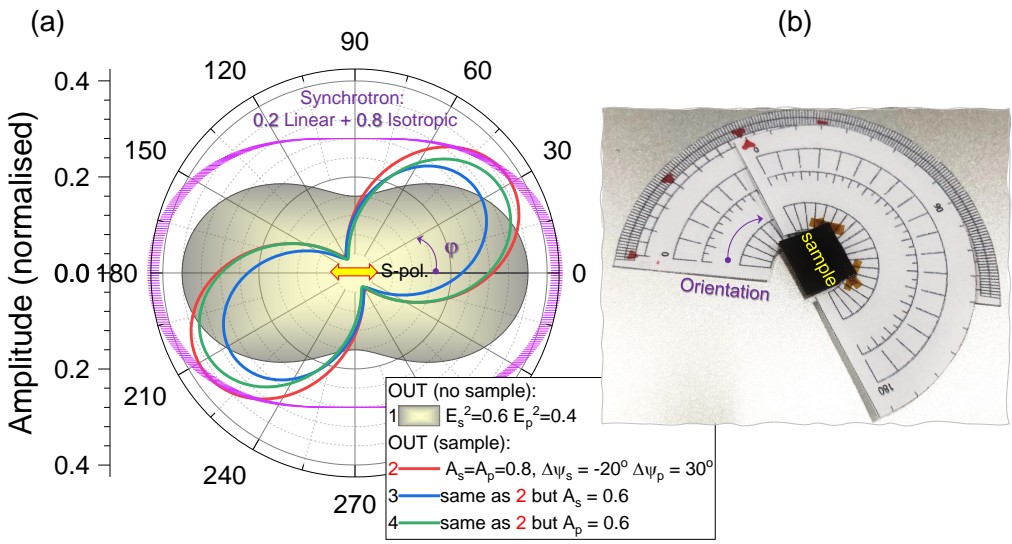

**Figure 6.** Numerical example of polarisation analysis at the ATR output (OUT) based on Equation (4). The angular dependence of synchrotron THz radiation (at THz beamline) comprised of 20% linear polarisation and 80% isotropic (circular) [28] is shown. (**b**) Sample holder for four-polarisation (4-Pol) ATR measurements: instead of changing the linear polarisation of light beam (THz), the sample orientation can be changed. Sample shown is isotropic THz absorber—black paper.

Figure 6 illustrates the use of Equation (4) for analysis of beam at the OUT-port of ATR. The analysis is based on two perpendicular field components, $E_s$ and $E_p$, with independent amplitudes. Isotropic and anisotropic absorbance changes are accounted for by the coefficients $A_{s,p}$. Changes of phases of the s-/p-components with the sample ($n_2$) placed on top of the ATR prism ($n_1$) are accounted for by the $\Delta\psi_{s,p}$. Isotropic absorbance in y- and z-orientations is presented by the curve *2* with $A_s = A_p = 0.8$ and phase changes corresponding to reflection from the low refractive index material (Figure A1b). Anisotropy along the y- or z-direction in the sample is modelled by a 20% change in coefficient $A_{s,p}$. As a result, the figure-8 (dipole-like absorbance) is rotated/tilted (an angular shift is introduced). This illustrates that polarisation analysis of the ATR results can be a useful tool for the measurement of anisotropy in the refractive index as well as absorbance by fitting Equation (4) with the experimentally measured data with the polarisation analysis at the OUT port. The TM-mode can experience strong field enhancement at the interface (Figure 4) and can cause additional absorption. Sub-wavelength structures at the sample-ATR prism interface cause depolarisation of the $E$-field components, e.g., $E_x = \sqrt{[E_s^2 + E_p^2] - (E_y^2 + E_z^2)}$ in the sample's frame of reference (Figure 5a); here, the intensity of the incident beam is $E_{in}^2 = E_s^2 + E_p^2$. The above described polarisation changes in the case of ATR should be possible to determine, using polarisation analysis at the OUT port and it will be carried out in future studies.

Equation (4) can be fitted by $Fit(\varphi) = Amp \times \cos(2\varphi + 2\phi) + Offset$, which is used for the 4-Pol method (see the synchrotron radiation angular distribution in Figure 6a, plotted with such dependence). The angle $2\phi$ indicates orientation of the sample (e.g., absorbing dipoles) and $Amp = (I_{max} - I_{min})/2$ and $Offset = (I_{max} + I_{min})/2$ with $I_{max,min}$, the maximum and minimum intensities of the transmitted or reflected light. By analysing the OUT-port polarisation and using the 4-Pol fit, it should be possible to establish anisotropy in absorbance ($A_{s,p}$) and refractive index ratio $\frac{n_2}{n_1}$ defining $\Delta\psi_{s,p}$. For large samples fitting on 3-mm-window of ATR, sample rotation can be used to realise the 4-Pol method (Figure 6b). Visualisation of the 4-Pol-ATR method (Figure 6) is extendable to the hyperspectral dimension since a broad THz spectrum is recorded. Metallic grid polarisers are polarisation insensitive and by measuring spectra at several angles with the analyser on the OUT port or by rotating the sample on the ATR prism, sample anisotropy can be examined. One can envisage an ATR geometry, where the entire ATR prism with a fixed sample can be rotated (around the z-axis (Figure 5)). Such measurements would require a hemispherical or conical ATR prism and could provide a more flexible setup, e.g., as it is realised for the synchrotron IR microscopy with the Ge-prism [30,31]. The realisation of the polarisation rotation on the sample stage is established in visible light polariscopy, where it is based on an electrically controlled optical phase retardance in a liquid crystal cell [32].

The high sensitivity ATR detection is realised with multi-reflection prisms, where the number of reflections at the sample–prism interface can occur over ten times before the light is detected [33]. Amplitude and phase changes for such a situation become cumulative, and the reflection (for the E-field) can be expressed as follows [34] (for ATR geometry):

$$\tilde{r}(\omega) = \frac{n_1 - [n(\omega) + ik(\omega)]}{n_1 + [n(\omega) + ik(\omega)]} \equiv r(\omega)e^{i\theta(\omega)}, \tag{5}$$

where $n_1$ is the refractive index of the ATR prism, $\tilde{n}(\omega) = n(\omega) + ik(\omega)$ is the complex index of the sample, $r(\omega) = \sqrt{R(\omega)}$ is the field reflection coefficient calculated from the experimentally measured reflectance $R$ spectrum, and $\theta$ is the phase change introduced by the ATR prism. The refractive index $\tilde{n} = n + ik$ of the sample is then the following [34]:

$$n = \frac{1 - r^2}{1 + r^2 + 2r\cos\theta} \times n_1; \quad k = \frac{-2r\sin\theta}{1 + r^2 + 2r\cos\theta} \times n_1, \tag{6}$$

and can be determined at each wavelength where the reflectivity $r(\omega)$ was measured, while the phase $\theta(\omega)$ is derived via Kramers–Kronig relation from $r(\omega)$. It is usually possible to calculate the reflectivity and phase changes from a single reflection for the entire number $N$ of reflections (see Equation (5)) as $\tilde{r}(\omega) = [r_{one}(\omega)]^N \times e^{iN\theta_{one}(\omega)}$, where $r_{one}$ and $\theta_{one}$ are the corresponding changes occurring at the single reflection [35].

### 2.5. Feasibility Test

The s- and p-pol. polarisation ratio of the synchrotron THz beam is not the same on the sample, due to multiple reflections caused by mirrors in the ATR unit (Figure 7a). The THz beam reflected from the sample–prism interface experiences further reflections until it reaches the OUT port. In total, from the IN to the OUT port, there are ten reflections, including one from the the sample. The beam inside the ATR unit is focused by parabolic mirrors and experiences a change in the s-/p-portions, due to the change in the direction of propagation inside the unit. Due to focusing onto the sample (a 3 mm window), a range of incidence angles are covered, which contributes to different Fresnel reflection coefficients and a change in the phase.

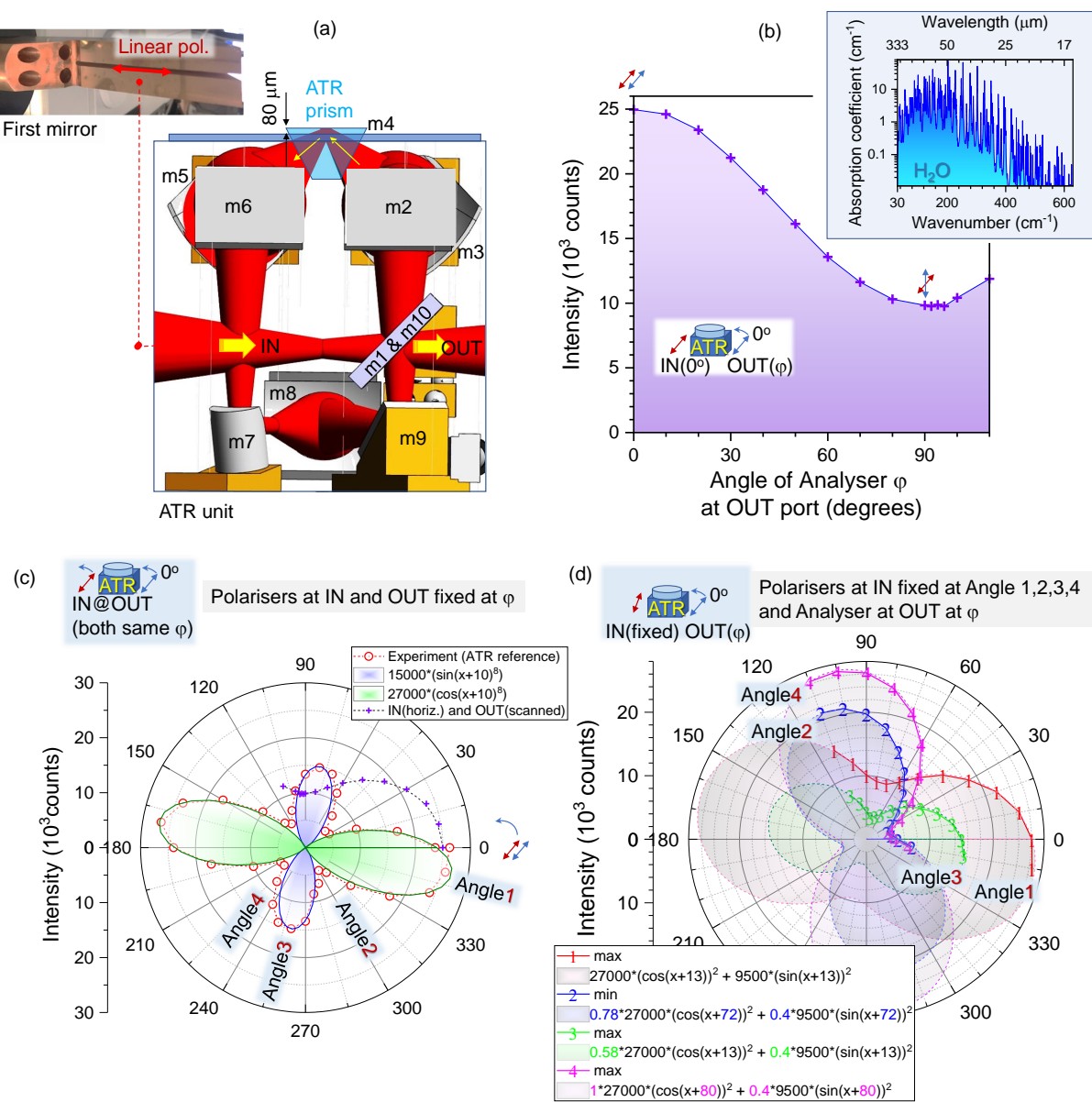

**Figure 7.** (**a**) Schematic of the ATR unit (Pike) used in this study. From the port IN to OUT, there are ten reflections (mirrors m1–10). (**b**) Cross-polarised performance of the ATR unit. The polarisation is set horizontal ($\varphi = 0°$) at the IN port. This corresponds to the maximum of linear polarisation of synchrotron radiation extracted by the first mirror (slotted mirror at the bending magnet of the synchrotron). THz spectrum over 30–630 cm$^{-1}$ (0.9–18.9 THz) is integrated (spectral range is defined by the mylar beamsplitter, and the detector is a Si bolometer). The water vapour absorption coefficient at 1 atm over the same spectral window [36] is shown in the inset. (**c**) Co-rotation of the mesh-grid polariser and analyser (at the IN and OUT ports): dots are experimental data and fits by $E_s^2 \propto \cos^8(\varphi + \Delta\psi_s)$ and $E_p^2 \propto \sin^8(\varphi + \Delta\psi_p)$ intensity components (see text for details). Four angles separated by 45° degrees are selected, where the local maxima or minima are observed (Angles 1–4). (**d**) Polarisation analysis of signal at port OUT for Angles 1–4 (at the IN port): experimental data by markers 1–4 and fit by Equation (4). The legend shows the fit functions explicitly.

The polarisation response of the ATR unit without sample (in air ambient) is analysed. A mesh grid polariser is set at the IN port and an analyser is set at the OUT-port. Linear (horizontal) polarisation is set to enter the ATR unit, which corresponds to the linear pol. of the synchrotron radiation ($\varphi = 0°$ in Figure 7b). A spectral window of 30–630 cm$^{-1}$ is selected with the beamsplitter, and the detector is a Si bolometer. A mesh-grid polariser/analyser has the same extinction ratio $E_r = \frac{T_{max}}{T_{min}}$ over the all wavelengths used; $T_{max,min}$ is the transmission at the maximum and minimum orientation of the polariser, respectively.

The measured ATR reference signal without a sample is only affected by the THz active absorbers/scatterers in air; hence, mostly water at the spectral window is used. The calculated water spectrum over this spectral window in shown in the inset of Figure 7b. The evanescent field extends $\sim\lambda/4$ into the air and experiences absorption and phase changes of s-/p-pol., corresponding to the refractive index ratio $n_2/n_1 \approx 1/2.42$ (without the sample with $n_2^{(air)} = 1$). The result of the change in polarisation between the s- and p-pol. and beam focusing (range of incidence angles) is a low extinction ratio for the entire unit $E_r \approx 2.5$. There is strong light leakage from the ATR unit, even in the crossed polariser–analyser conditions shown in Figure 7b; the $E_r$ of the mesh grid polarisers is $\sim 10^3$. The influence of the input polarisation and internal inter-change between the s- and p-pol. is revealed by co-rotating the mesh grid polarisers at the IN and OUT ports (Figure 7c). More angularly narrow s- and p-pol. lobes are observed, as compared to the extinction ratio measurements shown in Figure 7b and the cross vs circle markers in Figure 7c. There are two maxima and two minima per $\pi$-range of the $\varphi$ rotation, marked as Angles 1–4 in Figure 7c. The intensities of the components are fitted with $E_s^2 = Amp_s \times \cos^2(\varphi + \Delta\psi_s)$ and $E_p^2 = Amp_p \times \sin^2(\varphi + \Delta\psi_p)$ with phase angles set $\Delta\psi_s = \Delta\psi_p = 10°$. The value of the phase angle is larger than the possible alignment error of the polarisers, which is smaller than $\pm 5°$. Cumulative phase changes on the nine mirrors and the ATR prism (ten reflections) are the most probable cause of this phase change. Due to co-rotation of the polariser and analyser (on the IN and OUT ports, respectively) the change between s- and p-pol. becomes easily detected since such an alignment has the maximum transmission for the same polarisation, either s- or p-pol. Indeed, the fits plotted by the expressions given above miss the experimental results by a large margin (not shown in an already busy panel (c)). The solid lines in Figure 7c show a good fit when the $\cos^n(\varphi + \Delta\psi_s)$ and $\sin^n(\varphi + \Delta\psi_p)$ are used with $n = 8$, which corresponds to the number of reflections without two reflections from the flat M1 and M10 mirrors. Each reflection on the mirror contributes to the overall transmission $T_{IN-OUT} \propto R_{m2} \times R_{m3} \times ...R_{m9} \propto R^8$.

A perfect fit of the output signal (port OUT) can be obtained by the combined $E_s^2$ and $E_p^2$ intensity addition by Equation (4). The $A_{s,p}$ are the arbitrary amplitudes accounting for both the absorption (losses) and changes in amplitude due to reflection; $\Delta\phi_{s,p}$ are the phases for s- and p-polarisations, respectively. The angle $\varphi$ defines the orientation of the linear polarisation in the incident beam, i.e., for $\varphi = 0°$ only $E_s$ polarisation is present and $\varphi = 90°$ corresponds to pure $E_p$. When pure $E_s$ polarisation is incident on the sample and there are no changes in the phase of the reflected beam, the orientational output power will have a horizontally aligned figure-8; consequently, for the pure $E_p$ case, it will have a vertical figure-8. Figure 7d shows the polarisation analysis at the OUT port when the IN port polariser is set to one of the local min or max orientation angles (Angles 1–4 shown in Figure 7c). Changes in the amplitude and phase angles at four settings of Angles 1–4 are colour coded in the legend in Figure 7d. This shows that Equation (4) can be used for the arbitrary orientation of polarisation at the IN port.

Changes in amplitudes $A_{s,p}$ due to absorption in the sample and phase angle changes $\Delta\psi_{s,p}$ are due to refractive index contrast $n_2/n_1$ and can be accounted for using a fixed polarisation for excitation then carrying out polarisation analysis at the output. It is also obvious that such analysis has to be carried out for each spectral point.

### 2.6. Combined Spectral Filters and Polarisers

A toolbox of spectral filters and polarisers are required to fully explore the potential of spectroscopy for material characterisation. Such a toolbox is less developed for the THz spectral range. However, due to the low-resolution photolithography required for longer sub-millimetre wavelengths, spectral filters can be easily made based on a square lattice (Figure 8a). Here, we introduce the concept of filter multiplexing for control over the transmission wavelength, bandpass, and amplitude. Such comb filters can be made by direct laser cutting [37] from a metal foil or by standard photolithography. For the chosen wavelength $\lambda_{THz}$ [μm], the period $P = 0.78393 \times \lambda_{THz}$, the length of cross opening

$L = 0.50897 \times \lambda_{THz}$, and the width of the opening $W = 0.1482 \times \lambda_{THz}$ [38,39]. The typical 10–15% bandwidth $\Delta\lambda/\lambda_0$ for the geometry defined above becomes smaller as the $P/L$ and $P/W$ ratios are increased, while the centre frequency of the band-pass filter is mainly dependent on the cross-member length $L$. It is demonstrated that transmission $T > 0.9$ can be obtained using micrometer-thick Cu-foil within the 0.5–2 THz spectral range, using the presented scaling [38].

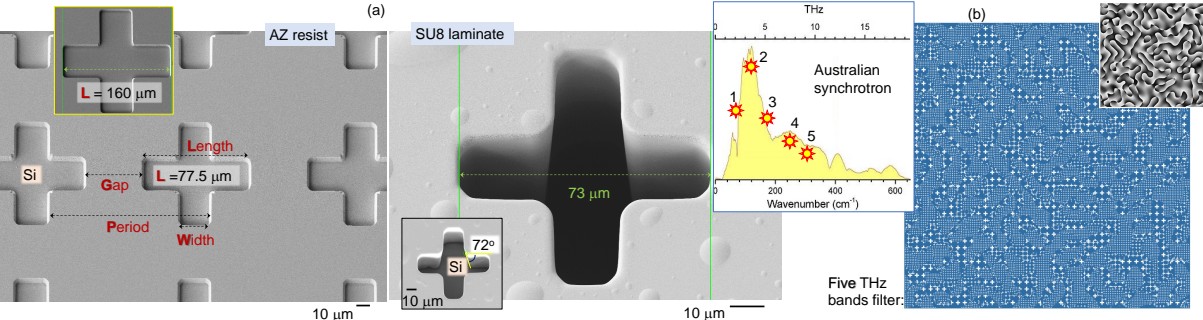

**Figure 8.** (**a**) SEM images of THz filters defined by photolithography: ~6 μm-thick AZ4562 resist (applied on surface) and SU8 laminate 20–500 μm thickness (SUEX from DJ Microlaminates Ltd.; applied or free-standing.) The period, length and width $P, L, W$ define the central wavelength $\lambda_0$ and the bandwidth $\Delta\lambda$ of the filter [38]; gap $G = P - L$. (**b**) Superimposed five THz bands filter based on the cross apertures (**a**). The binary mask (false colour) and the $(0 - 2\pi)$ phase map of the filter (see details in Figure 9). Spectrum of Australian synchrotron radiation at the THz beamline [40] is shown with selected five frequencies (star-markers).

With the known spectral profile of the synchrotron source (inset of Figure 8b), it is possible to design a transmission mask, which filters spectral bands and sets their intensities. Such filters are numerically designed to produce a comb of THz frequencies (five for the pattern shown in Figure 8b). As is known, it is desirable to have a uniform spectral response such that the gain of the bolometer can be precisely set in advance. The spectral profile can be precisely engineered by spatially multiplexing cross-filter designs corresponding to different wavenumbers. The first method is a super-pixel type spatial multiplexing, where every pixel of the mask is formed by the mosaic of the building block of the different filters. A super-pixel design with $5 \times 5$ elements is shown in Figure 9a. This is the most straightforward approach, but results in undesirable diffraction orders due to the periodic configuration. The second approach uses a random phase mask. In principle, any random phase mask will work, but in order to allow at least certain number of crosses of the same type, it is crucial to engineer the scattering ratio of the random phase mask. Such quasi-random phase masks can be synthesized using the well-known Gerchberg–Saxton algorithm (GSA) by giving the scattering ratio as the input [41]. The schematic of the GSA is shown in Figure 9b. The algorithm begins with a complex amplitude consisting of a uniform amplitude and random phase function at Plane 1, which is Fourier transformed. The amplitude in Plane 2 is replaced by a uniform amplitude within a predefined area, which decides the scattering degree given as $\sigma = d/D$, where $d$ is the length of the predefined area and $D$ is the length of the total area. The output of the GSA is a quasi-random phase mask, consisting of grey levels from 0 to $2\pi$. As the area of every different phase level is equal, different grey level ranges can be set to different filters. By controlling the relative ranges, the relative intensity responses can be manipulated. The images of a section of spatially randomly multiplexed cross-filters for $\sigma = 0.1$, $\sigma = 0.15$ and $\sigma = 0.2$ are shown in Figure 9c–e, respectively. The use of photolithography on thick 20–500 μm SU8 laminates (SUEX from DJ Microlaminates Ltd.) is particularly appealing, due to the possibility of retrieving free-standing mesh filters. With a gold (or any other metal) coating, this completes the filter. The SEM images of the mosaic type comb type and randomly multiplexed cross-filters on gold-coated SU8 films on silicon substrate are shown in Figure 9f,g, respectively.

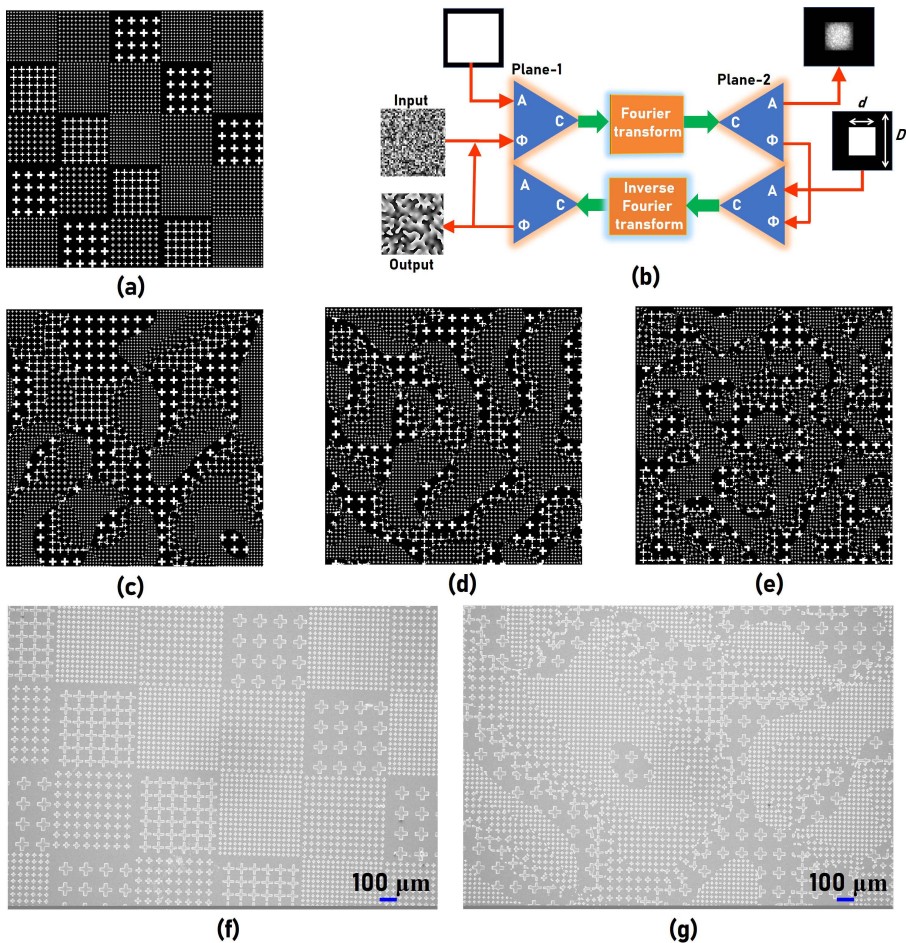

**Figure 9.** (**a**) Image of super-pixel mosaic design with $5 \times 5$ elements. (**b**) A block scheme of the Gerchberg–Saxton algorithm with scattering degree $\sigma = d/D$. Images of spatially randomly multiplexed cross-filters with for (**c**) $\sigma = 0.1$, (**d**) $\sigma = 0.15$ and (**e**) $\sigma = 0.2$. SEM images of the fabricated (**f**) mosaic type and (**g**) randomly-multiplexed cross filters on gold-coated SU8 films on silicon substrate.

Due to the isotropic geometry of the cross lattice and symmetry of the square lattice, an isotorpic polarisation response is expected [42]. Polarisation active filters can be fabricated using the same approach. Due to silicon's high transparency in the THz spectral band and readily available Si plasma etching for high aspect ratio structures, optical elements, such as optical vortex generators for spin and orbital angular momentum (SAM and OAM), manipulation at THz frequencies can be made [43]. A Fresnel rhomb with an apex angle $\sim 55°$ made from THz-transparent teflon or cyclic olefin copolymer TOPAS [44], which have a refractive index of $\sim 1.5$ over a wide THz spectral range can be used for a quarter waveplate generator of circular polarisation out of the linearly polarised input. This is especially useful for bio-medical and pharmaceutical applications where left- vs. right-hand enantiomers and chirally active molecules have to be detected.

Long wavelength (low frequency) optical radiation is promising for opto-mechanical applications [45–51]. Indeed, the reaction torque exerted onto an object by a beam of power $P$ is $\Gamma = \Delta\sigma P/\omega$, where $\Delta\sigma$ is the before–after change of the circular polarisation state [52].

## 3. Conclusions and Outlook

In this prospective/concept paper analysis of ATR measurements, we present most of the examples based on experiments at the THz beamline at the Australian Synchrotron. Explicit formulae pertinent to the ATR data analysis are presented for the discussion and illustration of practical examples as well as their tutorial value. Challenges in using THz

ATR units for the polarisation analysis are due to the large number of internal reflections: five to the ATR prism and an extra five to the output port. Moreover, focusing is carried out on the beam propagation to the sample, and those reflections have complex angular dependence and distribution of angles of incidence. This is additionally complicated by the complex polarisation composition of the synchrotron beam. However, for the unique spectral window at <3 THz, where broadband table top THz sources are not available for high brilliance/intensity, synchrotron polarisation is mostly linear (∼90%; along the slit of the first mirror used to extract THz radiation from the synchrotron) and can be utilised for THz polariscopy, which is still at the very early stage of development. Spectral filters for specific spectral bands can be made using photolithography on free-standing SU8 laminates as shown above. A metal coating of the developed SU8 micro-sheets can produce single frequency transmission filters as well as intensity and bandwidth comb filters.

The potential of polarisation analysis in the ATR mode and development of spectral/polarisation filters is presented by concept examples. In addition, polarisation optics for SAM and OAM manipulation and the analysis of THz radiation have to be developed for the interrogation of complex orientational structures, especially in biological/medical samples.

**Author Contributions:** Conceptualization, M.R., S.J. and J.M.; methodology, M.R., D.A., Z.V., A.W.W., S.J., J.M.; validation, M.R., S.H.N., V.A., Z.V. and J.M.; formal analysis, M.R., Z.V. and J.M.; investigation, S.L., J.H., T.K., D.A. and Z.V.; data curation, M.R., Z.V., S.J. and J.M.; writing—original draft preparation, S.J. and M.R.; writing—review and editing, all the authors; visualization, S.J. All authors have read and agreed to the published version of the manuscript.

**Funding:** This work was supported by JST CREST Grant Number JPMJCR19I3, Japan, and the ARC Discovery DP190103284 grants. The project was carried during EU16010, M15121 beamtimes at the Australian Synchrotron. Z.V. and A.W.W. are grateful for the support via the NHMRC grant 1042464. S.J. and S.H.N. are grateful for the support via the ARC Linkage LP190100505 project.

**Institutional Review Board Statement:** Not applicable.

**Informed Consent Statement:** Not applicable.

**Data Availability Statement:** The data presented in this study are available on request from the corresponding author.

**Conflicts of Interest:** The authors declare no conflict of interest.

## Appendix A. Phase and Amplitude Changes in Atr

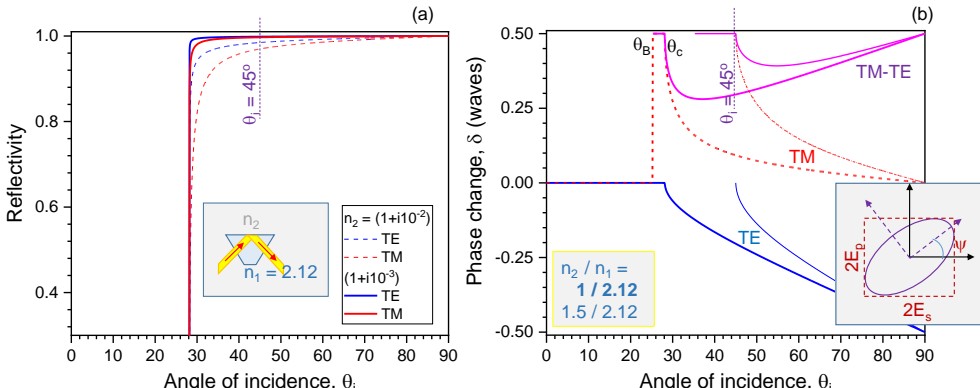

**Figure A1.** Visualisation of Equations (A1)–(A4). Reflection coefficients $R_{TE,TM}$ vs. angle of incidence $\theta_i$ for the ATR prism, e.g., $n_1 = 2.12$ at THz spectral range. Absorption in air is added by the imaginary part of refractive index $k_2 = 10^{-2}$ and $10^{-3}$. The inset shows ATR geometry and conventions. (**b**) The phase change upon reflection for the TE and TM modes governed by the real part of refractive indices $n = n_2/n_1$. The polarising Brewster angle is given by $\theta_B = \tan^{-1}(n)$ and the critical angle $\theta_c = \sin^{-1}(n)$. Thinner lines correspond to the case when the sample has a refractive index of $n_2 = 1.5$. The right inset in (**b**) shows the polarisation ellipse after the analyser.

The amplitude and phase of reflected light from the diamond prism interface with the sample changes (Equation (4)) and has to be accounted for in the analysis of light using the analyser. First, the changes without the sample (from ATR prism–air interface) are considered at different angle of incidence $\theta_i$. The reflectivity of the TM and TE modes is the following [53]:

$$\ln R_{TE}(\theta_i) = -\frac{4n_2^2 k_2}{n_1^2 \tan(\theta_i)\sqrt{1 - \frac{n_2^2}{n_1^2 \sin^2\theta_i}} \times \left(1 - \frac{n_2^2}{n_1^2}\right)}, \tag{A1}$$

$$\ln R_{TM}(\theta_i) = -\frac{4n_2^2\left(2 - \frac{n_2^2}{n_1^2 \sin^2\theta_i}\right)k_2}{n_1^2 \tan\theta_i \sqrt{1 - \frac{n_2^2}{n_1^2 \sin^2\theta_i}}\left(1 - \frac{n_2^2}{n_1^2 \sin^2\theta_i} + \frac{n_2^4}{n_1^4}\cot^2\theta_i\right)}, \tag{A2}$$

where $n_1 = 2.12$ is the refractive index of the ATR prism (medium 1) and $n_2 + ik_2$ (medium 2) is the sample (or air $n_2 = 1$) for the THz spectral range. Reflectivity at the interface changes the amplitude of the $E_{s,p}$ fields, which are detected by the analyser. Additionally, the phase changes upon ATR, depending on the angle, according to the Fresnel formulas defined for the relative refractive index $n = n_2/n_1$:

$$\phi_{TE}(\theta_i, n) = -2\tan^{-1}\left(\frac{\sqrt{\sin^2\theta_i - n^2}}{\cos\theta_i}\right), \tag{A3}$$

$$\phi_{TM}(\theta_i, n) = \pi - 2\tan^{-1}\left(\frac{\sqrt{\sin^2\theta_i - n^2}}{n^2\cos\theta_i}\right), \tag{A4}$$

for the $\theta_i > \theta_c$, where the critical angle $\theta_c = \sin^{-1}(n)$; the reflection is internal when $n < 1$. The evanescent field depth $1/\alpha$ is defined by the absorption coefficient $\alpha = \frac{2\pi}{k}\sqrt{\frac{\sin^2\theta_i}{n} - 1}$ (for $n < 1$).

The Fresnel formulas shown above are plotted in Figure A1. Reflectivity does not change the amplitude of the reflected TE and TM modes at angles of incidence larger than the critical angle $\theta_c$ (Figure A1a). Absorption of the THz beam due to the water humidity can be modelled by increasing the imaginary part of the refractive index $k_2$ (see solid vs. dashed lines in Figure A1a). The phase change upon reflection of TE and TM modes experiences complex changes at the polarising Brewster and critical angles (Figure A1b). Most importantly, the phase differences for the TM and TE modes experience the largest phase change close to the $\lambda/4$ condition for $\theta_i = 35 - 45°$. Typical ATR setups are designed for $\theta_i \approx \pi/4$; hence, a strong phase difference is expected, due to reflection from the sample–diamond interface, depending to the refractive index contrast $n$; see thinner lines in Figure A1b, which represent phase changes for $n = n_2/n_1 = 1.5/2.12$.

Upon reflection from the diamond–sample interface, both amplitudes of the $E_z$ (TM) and $E_y$ (TE) components can be changed because of the anisotropy of absorption, which is additionally affected by the phase change between the TM and TE modes, and is solely defined by the refractive index ratio at the interface. For the coherent E-field addition, the reflected light is elliptically polarised, and can be expressed in s-/p-components, measurable with an analyser (inset in Figure A1b). The ratio of s- and p-components of the E-field $\tan\beta = E_p/E_s$ and the phase differ upon reflection $\delta$. The polarisation ellipse is expressed by two angular parameters: the orientation angle $\psi'$ and the ellipticity angle $\chi'$ as $\tan(2\psi') = \tan(2\beta) \times \cos(\delta)$ and $\tan(2\chi') = \sin(2\beta) \times \sin(\delta)$. This polarisation change is mainly defined by the phase change $\delta$ upon reflection. For the incoherent light case, intensities of the two components $E_p^2$ and $E_s^2$ are added (Equation (4)).

### Appendix B. Experimental: Thz Beamline at Australian Synchrotoron

Some results obtained and ATR equipment discussed in this manuscript at at the THz/Far-IR beamline at the Australian Synchrotron (first light 2007). The polarisation of the synchroton radiation is a superposition of the linear (along the extraction mirror slit) and circular polarisations [28]. The two components originate from a constant and changing magnetic field through and at the entrance and exit of a bending magnet, i.e., the dipole- and edge-emissions, respectively. The THz/Far-IR beamline receives mainly the edge-emission, while the neighbouring IR microscopy beamline receives more of the dipole radiation. It is noteworthy that a bunch compression can be achieved to produce coherent synchrotron radiation (CSR) resulting in a flux increase of 2–3 orders in magnitude between 15 and 25 $cm^{-1}$ (0.45–0.75 THz; 667–400 mm) at the THz/Far-IR beamline.

Polarisation is defined as $E_s$ and $E_p$, which are $\perp$ and $\parallel$ to the plane of incidence (or TE and TM modes), respectively (Figure 1a). For anisotropic samples, those two components are absorbed differently, according to the alignment of the absorbing dipoles. Anisotropy of absorbance in the sample can be determined. The s-pol. (TE or $E_y$) probes the orientation of dipoles in the plane of the ATR window. Possibility to change the orientation of the sample is shown in Figure 6b. In the low refractive index $n_1 < 1.6$ (typical for bio-materials), the reflected light at a typical 45° incidence usually does not experience a phase change upon reflection at the sample–prism interface. However, the p-pol. (TM; $E_z$), which probes the absorbance in the direction perpendicular to the prism–sample interface (along the evanescent field) has a strong dependence of the reflected phase on $n_1$. At the $\theta_i = 45°$ incidence, a phase change of $\pi$ occurs when $\theta_B < \theta_i < \theta_c$; $\theta_B$ and $\theta_c$ are the Brewster and critical angle, respectively. The reflected $E_z$ component becomes $E_y$ at the output port. The consequence of the phase dependence of the TM mode makes the polarisation analysis of the ATR signals complicated, and most of the published data do not discriminate polarisation for excitation and do not carry out the polarisation analysis at the exit of the detector after the sample. Multi-reflection ATR units (Figure 2) introduce an uncertainty in the polarisation changes introduced by the sample and the rest of the setup.

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
