# Peer review of "Attenuated Total Reflection at THz Wavelengths: Prospective Use of Total Internal Reflection and Polariscopy"

_applsci, doi:10.3390/app11167632_

Round 1

Reviewer 1 Report

The work of Ryu et al. titled “Attenuated Total Reflection at THz wavelengths: peculiarities of total internal reflection and polariscopy” present a spectroscopic methodology for exploiting polarized ATR spectroscopy in the THz field for the study of matter. The topic described in the draft is extremely interesting, however, I didn't fully see the originality of the contents. For example, the first seven pages (out of 17) of the draft show data that were already published by the same authors in ref [6] and [7] ](Fig, A1 was already shown in [6], panel a of Figure 1 in [7], and Figure 3 is a combination of Figure 6 and Figure 7 in [7). In addition, the presentation of the 4-Pol methododology is not clearly written and more importantly, it is not corroborted by experimental data. For the mentioned reasons I can't recommend this contribution to be published in Applied Sciences.

Author Response

The work of Ryu et al. titled “Attenuated Total Reflection at THz wavelengths: peculiarities of total internal reflection and polariscopy” present a spectroscopic methodology for exploiting polarized ATR spectroscopy in the THz field for the study of matter. The topic described in the draft is extremely interesting, however, I didn't fully see the originality of the contents.

Answer. Thank you for the review remarks. This manuscript we chose in "concept paper" category where we combine recent results, as in review, and have added a concept to be tested in future experiments. The later is related to the Eqn 1.  

For example, the first seven pages (out of 17) of the draft show data that were already published by the same authors in ref [6] and [7] ](Fig, A1 was already shown in [6], panel a of Figure 1 in [7], and Figure 3 is a combination of Figure 6 and Figure 7 in [7).

Answer. Yes, we used material from our recent work where the novelty was to address the change of reflectivity of water and ice in bio-/medical samples. We changed text and adapted it for discussion to make it different from the published work. That part is now further re-edited to make material different with focus on new concept to investigate polarisation related effects.    

In addition, the presentation of the 4-Pol methododology is not clearly written and more importantly, it is not corroborted by experimental data. For the mentioned reasons I can't recommend this contribution to be published in Applied Sciences

Answer. Thank you for the comment. The 4-polarisation method is generic and works for all spectral ranges. This we presented  and it is rephrased to be better presented in the revised version.  The new concept in this manuscript is Eqn. 4 where X and Y-pol are added by intensity. We have preliminary tests which corroborated such polarisation analysis. Detailed experimental results will published in dedicated study. However, we agree that presented discussion in the manuscript is not direct transfer of 4-pol method to ATR geometry. This is now better presented in the revised version. Since this a concept/review  paper, we discuss much broader field of ATR applications and 4-pol method is only one part of it.   

Reviewer 2 Report

The authors are giving a conceptual system design based on the ATR limits and the total internal reflection limit by combining with polarization sensitive spectroscopy. As clearly addressed in the title:  “Attenuated Total Reflection at THz wavelengths: peculiarities of total internal reflection and Polariscopy”. Even if the given techniques are already well studied in the literature, the shown possibility of combing them for bio-matter investigations sound interesting and adds a sort of novelty.

I would like to thank all the Authors for their efforts and I kindly ask them to address my comments and suggestions below:

  1. The text is clearly written. The English language and style are fine still an overall spell would be beneficial.
  2. The techniques addressed in the title are all well studied in the literature, on the other hand the shown possibility of combing them for bio-matter investigations sound interesting and adds a sort of novelty. On the other hand the given manuscript is far away from being a prospective/concept paper. The target is not well defined or expressed. It is more like a short summary on previously done works and the strict correlation in between them is not clearly identified. To this content the logic sequence of the sections are also not well organized.

I kindly ask the authors to re-shape the manuscript to address the concept proposed in the title. The general idea must be better defined and expressed. The unnecessary sections must be cut and embedded in the concept related sections.

  1. In figure 1. “The index n = 1.6 also corresponds to the case when two n = 2.4 (diamond) slabs are separated by ngap = 1.5 (bio-sample), i.e., n/ngap = 1.6” The authors, take the crystal to the gap ratios, to estimate the mean index of refraction being sensed. I ask the authors to further explain with relevant references.
  1. The authors use the wave number[cm-1] in some figures while defining the spectral data in THz. I suggest the authors to use the THz frequencies only through out to text for an easier visualization to the readers.
  1. In line 248: “The phase change upon reflection of s-/p-polarizations (TE/TM modes) are solely defined by the real part of the refractive index.” The authors must clarify if this is their assumption or given as a common rule. In a common sense the polarization ratios are also dependent on the complex index of refraction. I kindly ask the authors to better clarify this concept.
  1. The expression given in equation 5 and the Kramers-Kronig relation given in equation 6 are valid under normal incidence. I kindly ask the authors to better explain the validity of the given solution for the case when multiple angular reflections are involved as in the case of an ATR system.
  1. The FDTD calculations/simulations given are at IR regime and do not correspond to the proposed concept of the article. I kindly ask the authors to add simulative results at the spectral regime of interest. The given meta-material structures( THz filters) are neither well characterized nor the purpose and/or correlation to the manuscript title is well expressed. The unnecessary sections must be cut and embedded in the concept related sections.
  1. As in comment 7: The same for the Combined spectral filters and polarisers. The unnecessary sections must be cut and embedded in the concept related sections.
  1. In line 353: “Debye rotation, network stretching, rocking and wagging vibrations of water molecules all are active at low-THz spectral band” this already a well studied concept. On the other hand this phrase has no place in the conclusion since the given examples nor the concept does not support the argument.
  2. I kindly ask the authors to avoid citing phrases in the conclusion. All arguments must be added in the scientific explanation parts throughout the text where needed.
  1. In line 365: “Indeed, the reaction torque exerted onto an object by a beam of power … circular polarization state [38]”. No place in the conclusion, I kindly ask the authors to place the expression within the scientific explanation part where needed.

Author Response

The authors are giving a conceptual system design based on the ATR limits and the total internal reflection limit by combining with polarization sensitive spectroscopy. As clearly addressed in the title:  “Attenuated Total Reflection at THz wavelengths: peculiarities of total internal reflection and Polariscopy”. Even if the given techniques are already well studied in the literature, the shown possibility of combing them for bio-matter investigations sound interesting and adds a sort of novelty.

I would like to thank all the Authors for their efforts and I kindly ask them to address my comments and suggestions below:

  1. The text is clearly written. The English language and style are fine still an overall spell would be beneficial.

Answer. Proofreading was made by a native English speaker. 

  1. The techniques addressed in the title are all well studied in the literature, on the other hand the shown possibility of combing them for bio-matter investigations sound interesting and adds a sort of novelty. On the other hand the given manuscript is far away from being a prospective/concept paper. The target is not well defined or expressed. It is more like a short summary on previously done works and the strict correlation in between them is not clearly identified. To this content the logic sequence of the sections are also not well organized.

I kindly ask the authors to re-shape the manuscript to address the concept proposed in the title. The general idea must be better defined and expressed. The unnecessary sections must be cut and embedded in the concept related sections.

Answer. Thank you for the valid remark. We reedited structure to address the concept vs former results proportions. Modification of title, abstract and entire manuscript were made. One uniting feature of this manuscript is a connection of generic concepts with experiments at the Australian synchrotron. Since experimental work does not pertain to this concept manuscript, experimental section is moved to appendix.   

  1. In figure 1. “The index n = 1.6 also corresponds to the case when two n = 2.4 (diamond) slabs are separated by ngap = 1.5 (bio-sample), i.e., n/ngap = 1.6” The authors, take the crystal to the gap ratios, to estimate the mean index of refraction being sensed. I ask the authors to further explain with relevant references.

Answer. Explanation is added in discussion of fig. 1.

  1. The authors use the wave number[cm-1] in some figures while defining the spectral data in THz. I suggest the authors to use the THz frequencies only through out to text for an easier visualization to the readers.

Answer. We agree that wavenumbers and THz might be confusing, however, use of them both is common in literature. They both scale as energy. We also explicitly write THz and wavelength for reference in the text (they are more confusing since are reciprocal). Wavenumber and  THz axes are added in Fig. 3.

  1. In line 248: “The phase change upon reflection of s-/p-polarizations (TE/TM modes) are solely defined by the real part of the refractive index.” The authors must clarify if this is their assumption or given as a common rule. In a common sense the polarization ratios are also dependent on the complex index of refraction. I kindly ask the authors to better clarify this concept.

Answer. Yes, it is correct, the complex refractive index is defining the phase difference between s and p pols. However, the main change is due to real part of the index. The effect of the imaginary part is smaller as shown in figure in appendix. Clarification is added in the text. The discussion of this matter was already in middle of the polarisation section 2.4. 

  1. The expression given in equation 5 and the Kramers-Kronig relation given in equation 6 are valid under normal incidence. I kindly ask the authors to better explain the validity of the given solution for the case when multiple angular reflections are involved as in the case of an ATR system.

Answer.  This is very good question/remark. We have made first data analysis of polarisaiton changes in the 10-reflection ATR unit we used (fig. 2). There is a way to present all changes in this complex unit acting as a wavelplate. This can be achieved  by plotting  transmission vs angular difference between polariser and analysier.  The detailed experimental manuscript is under preparation. We would not like to expand into technical details in this already long concept paper. 

If we can account polarisation  changes in transmission  through ATR unit as through a waveplate, we expect that we can use KK relations which are indeed usually discussed for  the normal incidence. However, the eqns 5-6 are already for the ATR geometry. Since KK relation is based on material response to perturbation, the non-normal incidence can be accounted by amplitude and phase changes readily available from the Fresnel formulas.   

  1. The FDTD calculations/simulations given are at IR regime and do not correspond to the proposed concept of the article. I kindly ask the authors to add simulative results at the spectral regime of interest. The given meta-material structures( THz filters) are neither well characterized nor the purpose and/or correlation to the manuscript title is well expressed. The unnecessary sections must be cut and embedded in the concept related sections.

Answer. With due respect, we disagree that FDTD calculations are not applicable to the presented discussion. In this concept manuscript we present idea to use Maxwell scaling to provide insights into long-wavelength response of material. Maxwell scaling is used to predict optical response of photonic crystals for changed period and refractive indices. Here we use it for THz instead of visible wavelengths. Related to the discussed concept of frustrated-TIR, the local field enhancement due to inclusions can strongly affect the light penetration depth (more light losses will take place in the regions of high intensity). We believe this is very important for mm-wavelength interaction with materials due to less expected E-field enhancement, which can also cause local heating.   

  1. As in comment 7: The same for the Combined spectral filters and polarisers. The unnecessary sections must be cut and embedded in the concept related sections.

Answer. This section is separated since it has its own technical merit and provides a concept of creating spectral "comb" filters with tailored transmission intensity. It can be very useful for spectrally complex THz sources as synchrotrons. Its is a concept, we show preliminary results of fabrication. This is why we strongly motivated to keep it as a separate section. Splitting larger text into smaller sections helps readability.   

  1. In line 353: “Debye rotation, network stretching, rocking and wagging vibrations of water molecules all are active at low-THz spectral band” this already a well studied concept. On the other hand this phrase has no place in the conclusion since the given examples nor the concept does not support the argument.

Answer. Thank you for the valid comment. We added discussion on importance of rotational-vibrational spectral window monitoring during phase transition, e.g., ice-water. Revisions are made.  

  1. I kindly ask the authors to avoid citing phrases in the conclusion. All arguments must be added in the scientific explanation parts throughout the text where needed.

Answer. We agree that it is usual not to use citations in the conclusions. This sections is "Conclusions and Outlook". This is way we repeated some citations which are important for future development of THz spectroscopy at low wavenumbers.  References removed from conclusions.

  1. In line 365: “Indeed, the reaction torque exerted onto an object by a beam of power … circular polarization state [38]”. No place in the conclusion, I kindly ask the authors to place the expression within the scientific explanation part where needed.

Answer. Revised.

Reviewer 3 Report

The paper submitted by Ryu and colleagues is a prospective paper about attenuated total reflection at terahertz frequencies. The paper is overall interesting as it lists the features, the applications and the challenges of total internal reflection and polariscopy. The information contained within the different section is -in my opinion- of great interest to the readers that would like to pursue researches in that specific field. Obviously, the content is not new since it is a concept paper.

There are a lot of terms that are english-style spelled, for instance "polarisation, characterisation..." but should rather be american-spelled.

In a nuthsell I would encourage to publish the paper, since the reported information would be beneficial for the community.

Author Response

Thank you for the positive evaluation of our concept paper manuscript. We use English-style spelling as it is acceptable in the journal. If editorial requirement will be placed to make corrections we will make a thorough style change.   

Round 2

Reviewer 1 Report

I thank the authors for submitting a revised version of the draft and replying to my comments. However, I am still not convinced about the current form of the paper that was only in small part modified. I reckon that a robust correction of all the mentioned flaws should be performed in order to improve the quality of the draft.

To my advice, the extent with which results from previous works are presented is yet not appropriate and the “concept” the authors want to present on the improvements offered by polarization studies to the technique is not yet in the focus of the paper.

Even if is now clear to me what the aim of the work was, the draft in the revised version does not send, to my advise, the message the authors want to send. I suggest again a robust rewriting of the paper, with wide cut of the experimental parts (first 7 pages). Moreover, also the english should be revised and typos corrected.

Moreover, there is a growing community working on THZ and sub THz region on biomaterials and biology, the authors should include some of the most relevant papers in the references.

Indeed, the bibliography is not appropriate, especially considering the amount of information contained in the paper, most of them not justified by literature data.

The basics of the ATR technique should be better presented and explained. Also the statements on spectroscopic details should be written with appropriate vocabulary in order to avoid misunderstandings. Below, I made a list of some details on more speficic and evident issues and some little english mistakes I found in the main text that need to be corrected.

- page 1 line 1 : “Capabilities of the Attenuated Total Reflection (ATR) at THz wavelengths for increased sub-surface depth characterisation of (bio-)materials is presentedhere the subject of the sentence is “Capabilities”, thus the verb should be “are presented”

  • page 2, line 32 : “There is a particular advantage in using Attenuted Total Reflection technique at Thz wavelenghts, where water absorption can be detrimental to free space THz beam propagation”. Here the sentece is not clear, as an IR spectroscopist I understand that this consideration is made by comparing ATR to transmission measurements, however, this is my deduction and it is not what is written in the text, where, besides, ATR spectroscopy has not yet been introduced. Moreover, literature data shown that THz transmission measurements on biomolecules and solutions can also be made in sealed cells under vacuum conditions. (Piccirilli et al. Biophysical chemistry,199,,17-24,2015).

  • page 2, line 39: “can be extended to an anvil or pressure cell (using diamond windows) experiments with materials at high pressure and temperature, recently demonstrated with a 35 MPa cell [1]. The reference cited here is not appropriate since the cited work is referred specifically to a setup optimized for sub Thz frequencies, please add also : Knake et al. , Review of Scientific Instruments 87, 104101 (2016)

  • page 2, line 45: “Due to both a large refractive index change from the water-ice transition ( 0.4) and change in absorption coefficient through the THz range, the ATR condition can be not fulfilled” here the authors refer to the ATR condition without mentioning it before.

  • Page 2, line 48: “there is a good refractive index contrast between liquid water and ice. The meaniing of the statement is not clear, why having a contrast between liquid water and ice is a good thing for ATR?

  • Page 2, line 53; “In this perspective/concept paper, based on recent experiments at the THz beamline at the Australian Synchrotron, technical peculiarities of the ATR operation mode are detailed.” here the authors state that the concept paper is based on recent experiments. This let the readers think that the focus of the paper is on the experimental results they present in the next 7 pages. If I understand well from the authors answer to the first referee round, the focal point of the paper is instead the 4-Pol technique that they present as an idea to be developed.

  • Pag 2 line 58: “Polarisation, intensity and spectral filtering of complex synchrotron THz radiation can provide more flexibility in material characterisation In the paragraphs before the authors assess that most of the issues in performing ATR THz spectroscopy are due to the refractive index dependence on frequency and on having a good contact of the sample with the ATR crystal. How spectral filtering can help avoiding these issues? And why THz radiation should be complex? Please explain

  • Pag 2 line 61: The ATR condition has not been defined before

  • Pag 2 line 62; “or due to light tunneling through a small air gaps between the surface of the ATR prism and sample. Basic principles of ATR should be introduced before talking about tunneling of light. Please also check the english.

  • Page 3, line 1: “Recent results based on measurements using synchrotron-ATR at the Australian Synchrotron are presented and the concept of polarisation discriminated measurements is outlined.Again, the focus seems to be on already published results and polarisation topic appears more as a conclusion.

  • Pag 3, line 69: The technique of attenuated total reflection (ATR) spectroscopy has become a popular method to measure dielectric properties of materials.ATR spectroscopy has been used in the last three decades for the characterization of materials in several different fields, in MID infrared it is a well extablished technique often used for quick screening of samples. Assessing that it has become a popular method without adding any reference is reductive and can led to misunderstandings. Please add some references: (Chittur et al. Biomaterials 1998 Mar;19(4-5):357-69, Piccirilli et al. Nanomaterials,11,,1103,2021,MDPI)

  • Pag 3, line 69: “It relies on the reduction of the reflected signal intensity at the total reflection angle, when there is no transmitted light into the sample (placed on top of the ATR prism). The technique relies on the incident energy being absorbed only via an evanescent wave generated at the interface between the sample and ATR prism. The advantage of the method lies in the fact that solid and liquid samples can be studied with minimal preparation.” Here the explanation of ATR functioning principles is not clear. This part should be to my advise rewritten.

  • Pag. 3 line 69: “This suggests that the ATR technique is sampling the absorption coefficient of the sample at the given frequency.This sentence has no point to be in the text, ATR technique is widely used over 30 decades and it is well known that there is a direct link between ATR signal and Absorption.

  • Pag. 3 line 74: “where the dp is of the order of 1 mm for biological tissue. At these distances, the total absorption of the evanescent wave at distances in the order of the dp is, indeed, negligible...(Fig.1):” In the mentioned paragraphs there are several inaccuracies that make the text barely understundable. For example: the discussion about the penetration depth appears confused. Indeed it is stated that the depth is 1 mm for biological tissues without specifying with what kind ATR crystal this rough evaluation is done. Second, it is stated that the absorption of the evanescent wave becomes not negligible at such depths in biological tissues, I would expect to read a motivation for the statement or at a couple of references. Last, when the authros refer to the transmission across “the gap of width d”. It is not clear what they are referring to. Please specify.

  • Pag. 3, line 86 : “...and light tunnels through the separation with lower refractive index.” When describing ATR method the authors never mention quantum light tunneling as driving pheomenon.

  • Pag. 3 line 90: “frustrated reflection mode” please, explain in few words what the frustrated reflection is.

  • Pag. 3 line 106: “...since the working surface of ATR prismWhat “the working surface of ATR prism” is? Mayde the authors refer to the sampling surfae of the crystal. Please Clarify in the main text.

  • Pag.5 line 122: “The penetration depth into sample at ATR conditions is sub-wavelength and the s-pol.s-pol is never defined

  • Pag. 6 line 133: what is Lurpak butter? Why the authors need to specify thaht it was Lurpak butter and not just butter? Is it a specific compounds or is it butter from a specific brand? Please explain

  • Pag. 6 line 134-146. “The temperature was varied from -20 to 24C over a total time frame of 24 min with continual scanning; a total of 140 sets of 40 averaged scans were collected during that time span. Figure 3 shows reflectance evolution with temperature at two spectral regions. The inset presents reflectivity changeswith temperature. There is a featureless temperature dependent reduction variation with butter which is evident at the most sensitive frequency range of 0.95 to 1.0 THz. Considerably more strong temperature dependence of reflectivity was observed at 2 THz band. Overall, the reflected ATR signal for Lurpak butter showed a 13% decrease for over the -20C to +24C change at 1.0 Thz and a 3.5 % decrease at 2 THz. A slight decrease in reflectance with temperature variation 20C around water freezing is indicating that the water content 14.7% needs to be regarded as being “bound” in a homogenous mixture of fat and protein. The temperature related reduction in reflectance in the region of 2.0 THz showed a plateau in the 2-12C range. It may be related to a transition in the constituent fats. These results would suggest that other materials with bound water may not show the effect of water freezing.these results are alredy commented in another paper, The authors should just mention the paper and summarize the results instead of writing a whole paragraph on them. In the paper of “Lundholm et al. RSC advances,4,49,25502-25509,2014” an experiment performed on proteins in solution with THz transmission under N2 purged atmosphere is reported and consideration about absorbance changes in temperature are also made. The authors should include it to the references (possibly togheter with other similar experiments) and correct all the technical inaccuracies in the text.

  • Pag.6 line 168: “The experimentally determined polarisation was 90% linear and 10% circular due to sum of the linear and circular polarisations at < 100 cm-1.” polarization of what? Is it referred to the SR beam? Please specify and add references about the IR beam characteristics at the australian synchrotron

  • Pag 7 line 178: “Intensity distribution in the case of frustrated TIR depends...” TIR is not defined before. Please add references

  • Pag 8 line 179: “The wavelength scaling in frustrated TIR gives insights into light tunneling regardless of the wavelength.” The authors never say what tunneling is, lease specify and add references

  • at pag 11 line 285 (and following) “The high sensitivity ATR detection is realised with multi-reflection slabs/prisms where the number of reflections from the sample-prism interface can increase more than ten times [25].” the sentence here is not clear, please explain

  • at pag 12: “A toolbox of spectral filters and polarisers are required to fully explore the potential of spectroscopy for material characterisation. Such a toolbox is less developed for the THz spectral range.” less developed compared to what spectral range? can the author explain why?

  • At pag 12: “However, due to low resolution photo-lithography required for longer sub-mm wavelengths, spectral filters can be easily made based on a square lattice” The authors state that the filter can be easly made, do they prepare themself these filters or are them commercially available? This detail should be specified and references should be added (if available).

  • As is known, it is desirable to have a uniform spectral response such that the gain of the bolometer can be precisely set in advance.” The bolometer is mentioned here for the first time, The authors should specify what a bolometer is or at least say before in the text that they used a bolometer as a detector.

  • Pag 13: “cross-filter designs corresponding to different wavenumbers. The first method is a super-pixel type spatial multiplexing where every pixel of the mask is formed by the mosaic of the building block of the different filters. A super-pixel design with 5  5 elements is shown in Fig. 8(a). This is the most straightforward approach but results in undesirable diffraction orders due to the periodic configuration. The second approach uses a random phase mask. In principle, any random phase mask will work but in order to allow at least certain number of crosses of the same type, it is crucial to engineer the scattering ratio of the random phase mask. Such” here there is a total lack of references

  • Page 13 line 344: “However, for the unique spectral window at < 3 THz where table top THz sources are not available for high brilliance/intensity,Thie statement is not precise, there exist Thz laser sources delivering light below 3 Thz.

  • In the Figures caption and in the main text it is often found the plot of “Reflectance vs frequency”. I have two comments about it. First, the authors should show the frequencies always with same units (ether Thz or cm-1), second, is reflectance meant for what is commonly called ATR-absorption? If yes, the authors should clearly explain why they call it reflectance by clearly referring to the experimental setup used.

Author Response

thank you for the critical remarks. they are answered below. 

---------------------------------------------------------------------------------

 However, 

However, I am still not convinced about the current form of the paper that was only in small part modified. I reckon that a robust correction of all the mentioned flaws should be performed in order to improve the quality of the draft.

Answer. we have added section 2.5 with fig 7 as an experimental proof of the validity of eqn.4. 

To my advice, the extent with which results from previous works are presented is yet not appropriate and the “concept” the authors want to present on the improvements offered by polarization studies to the technique is not yet in the focus of the paper.

Answer. section 2.5 is added to address polarisation issue. 

Even if is now clear to me what the aim of the work was, the draft in the revised version does not send, to my advise, the message the authors want to send. I suggest again a robust rewriting of the paper, with wide cut of the experimental parts (first 7 pages). Moreover, also the english should be revised and typos corrected.

Answer. Experimental part is taken to supplement. The first sections address situation when ATR conditions fails. It shows how to use it. Manuscript is English proofread by a native speaker.  

Moreover, there is a growing community working on THZ and sub THz region on biomaterials and biology, the authors should include some of the most relevant papers in the references. Indeed, the bibliography is not appropriate, especially considering the amount of information contained in the paper, most of them not justified by literature data.

Answer. Discussion expanded, references added.

The basics of the ATR technique should be better presented and explained. Also the statements on spectroscopic details should be written with appropriate vocabulary in order to avoid misunderstandings. Below, I made a list of some details on more speficic and evident issues and some little english mistakes I found in the main text that need to be corrected.

Answer. Mended.

- page 1 line 1 : “Capabilities of the Attenuated Total Reflection (ATR) at THz wavelengths for increased sub-surface depth characterisation of (bio-)materials is presented” here the subject of the sentence is “Capabilities”, thus the verb should be “are presented”

Answer.  Thank you. corrected. 

  • page 2, line 32 : “There is a particular advantage in using Attenuted Total Reflection technique at Thz wavelenghts, where water absorption can be detrimental to free space THz beam propagation”. Here the sentece is not clear, as an IR spectroscopist I understand that this consideration is made by comparing ATR to transmission measurements, however, this is my deduction and it is not what is written in the text, where, besides, ATR spectroscopy has not yet been introduced. Moreover, literature data shown that THz transmission measurements on biomolecules and solutions can also be made in sealed cells under vacuum conditions. (Piccirilli et al. Biophysical chemistry,199,,17-24,2015).

Answer.  Thank you. corrected. 

  • page 2, line 39: “can be extended to an anvil or pressure cell (using diamond windows) experiments with materials at high pressure and temperature, recently demonstrated with a 35 MPa cell [1]. ” The reference cited here is not appropriate since the cited work is referred specifically to a setup optimized for sub Thz frequencies, please add also : Knake et al. , Review of Scientific Instruments 87, 104101 (2016)

Answer.  Thank you for good reference. Rephrased. 

  • page 2, line 45: “Due to both a large refractive index change from the water-ice transition ( 0.4) and change in absorption coefficient through the THz range, the ATR condition can be not fulfilled” here the authors refer to the ATR condition without mentioning it before.

Answer.   Total reflection is known and it is obvious from the context of discussion.  Rephrased, explanation added.

  • Page 2, line 48: “there is a good refractive index contrast between liquid water and ice.” The meaniing of the statement is not clear, why having a contrast between liquid water and ice is a good thing for ATR?

Answer. Rephrased. Formation of ice invalidates ATR, but it is shown how to make use of this condition.

  • Page 2, line 53; “In this perspective/concept paper, based on recent experiments at the THz beamline at the Australian Synchrotron, technical peculiarities of the ATR operation mode are detailed.” here the authors state that the concept paper is based on recent experiments. This let the readers think that the focus of the paper is on the experimental results they present in the next 7 pages. If I understand well from the authors answer to the first referee round, the focal point of the paper is instead the 4-Pol technique that they present as an idea to be developed.

Answer. Addition of section 2,5 and fig 7 resolves this issue. 

  • Pag 2 line 58: “Polarisation, intensity and spectral filtering of complex synchrotron THz radiation can provide more flexibility in material characterisation” In the paragraphs before the authors assess that most of the issues in performing ATR THz spectroscopy are due to the refractive index dependence on frequency and on having a good contact of the sample with the ATR crystal. How spectral filtering can help avoiding these issues? And why THz radiation should be complex? Please explain
    Answer. Synchrotron radiation has combination of linear and circular polarisaitons with different ratios at different wavelengths. This we call complex. Spectral window of relevant measurements is important. This what we meant by listing different issues discussed in the manuscript. 
  • Pag 2 line 61: The ATR condition has not been defined before
  • Answer. Now defined. 
  • Pag 2 line 62; “or due to light tunneling through a small air gaps between the surface of the ATR prism and sample.” Basic principles of ATR should be introduced before talking about tunneling of light. Please also check the english.
  • Answer. Now defined. 
  • Page 3, line 1: “Recent results based on measurements using synchrotron-ATR at the Australian Synchrotron are presented and the concept of polarisation discriminated measurements is outlined.” Again, the focus seems to be on already published results and polarisation topic appears more as a conclusion.
  • Answer. Solved by sec 2.5. 
  • Pag 3, line 69: “The technique of attenuated total reflection (ATR) spectroscopy has become a popular method to measure dielectric properties of materials.” ATR spectroscopy has been used in the last three decades for the characterization of materials in several different fields, in MID infrared it is a well extablished technique often used for quick screening of samples. Assessing that it has become a popular method without adding any reference is reductive and can led to misunderstandings. Please add some references: (Chittur et al. Biomaterials 1998 Mar;19(4-5):357-69, Piccirilli et al. Nanomaterials,11,,1103,2021,MDPI)
  • Answer. Added.
  • Pag 3, line 69: “It relies on the reduction of the reflected signal intensity at the total reflection angle, when there is no transmitted light into the sample (placed on top of the ATR prism). The technique relies on the incident energy being absorbed only via an evanescent wave generated at the interface between the sample and ATR prism. The advantage of the method lies in the fact that solid and liquid samples can be studied with minimal preparation.” Here the explanation of ATR functioning principles is not clear. This part should be to my advise rewritten.
  • Answer. Rephrased.
  • Pag. 3 line 69: “This suggests that the ATR technique is sampling the absorption coefficient of the sample at the given frequency.” This sentence has no point to be in the text, ATR technique is widely used over 30 decades and it is well known that there is a direct link between ATR signal and Absorption.
  • Answer. Rephrased. 
  • Pag. 3 line 74: “where the dp is of the order of 1 mm for biological tissue. At these distances, the total absorption of the evanescent wave at distances in the order of the dp is, indeed, negligible...(Fig.1):” In the mentioned paragraphs there are several inaccuracies that make the text barely understundable. For example: the discussion about the penetration depth appears confused. Indeed it is stated that the depth is 1 mm for biological tissues without specifying with what kind ATR crystal this rough evaluation is done. Second, it is stated that the absorption of the evanescent wave becomes not negligible at such depths in biological tissues, I would expect to read a motivation for the statement or at a couple of references. Last, when the authros refer to the transmission across “the gap of width d”. It is not clear what they are referring to. Please specify.
  • Answer. Rephrased. When there is a gap at interface, light can tunnel through for the conditions discussed in fig. 1b
  • Pag. 3, line 86 : “...and light tunnels through the separation with lower refractive index.” When describing ATR method the authors never mention quantum light tunneling as driving pheomenon.
  • Answer. Tunneling is discussed in frustrated TIR (formulas 2 and Fig 1b). Rephrased.
  • Pag. 3 line 90: “frustrated reflection mode” please, explain in few words what the frustrated reflection is.
  • Answer. Tunneling of light through the gap is frustrated TIR (formulas 2 and Fig 1b). 
  • Pag. 3 line 106: “...since the working surface of ATR prism” What “the working surface of ATR prism” is? Mayde the authors refer to the sampling surfae of the crystal. Please Clarify in the main text.
  • Answer. It is the interface between the ATR prism and the sample. Removed working.
  • Pag.5 line 122: “The penetration depth into sample at ATR conditions is sub-wavelength and the s-pol.” s-pol is never defined
  • Answer. s and p-pol. are two generic polarisations. Defined now.
  • Pag. 6 line 133: what is Lurpak butter? Why the authors need to specify thaht it was Lurpak butter and not just butter? Is it a specific compounds or is it butter from a specific brand? Please explain
  • Answer. It is a butter brand.  
  • Pag. 6 line 134-146. “The temperature was varied from -20 to 24 C over a total time frame of 24 min with continual scanning; a total of 140 sets of 40 averaged scans were collected during that time span. Figure 3 shows reflectance evolution with temperature at two spectral regions. The inset presents reflectivity changeswith temperature. There is a featureless temperature dependent reduction variation with butter which is evident at the most sensitive frequency range of 0.95 to 1.0 THz. Considerably more strong temperature dependence of reflectivity was observed at 2 THz band. Overall, the reflected ATR signal for Lurpak butter showed a 13% decrease for over the -20 C to +24 C change at 1.0 Thz and a 3.5 % decrease at 2 THz. A slight decrease in reflectance with temperature variation  20 C around water freezing is indicating that the water content  14.7% needs to be regarded as being “bound” in a homogenous mixture of fat and protein. The temperature related reduction in reflectance in the region of 2.0 THz showed a plateau in the 2-12 C range. It may be related to a transition in the constituent fats. These results would suggest that other materials with bound water may not show the effect of water freezing.” these results are alredy commented in another paper, The authors should just mention the paper and summarize the results instead of writing a whole paragraph on them. In the paper of “Lundholm et al. RSC advances,4,49,25502-25509,2014” an experiment performed on proteins in solution with THz transmission under N2 purged atmosphere is reported and consideration about absorbance changes in temperature are also made. The authors should include it to the references (possibly togheter with other similar experiments) and correct all the technical inaccuracies in the text.
  • Answer. This is an example section. Text was shortened. Figure is different from the published one. Discussion is added in introduction. 
  • Pag.6 line 168: “The experimentally determined polarisation was 90% linear and 10% circular due to sum of the linear and circular polarisations at < 100 cm-1.” polarization of what? Is it referred to the SR beam? Please specify and add references about the IR beam characteristics at the australian synchrotron
  • Answer. Polarisation of AuSy at THz beamline. 
  • Pag 7 line 178: “Intensity distribution in the case of frustrated TIR depends...” TIR is not defined before. Please add references
  • Answer. Was defined on page 3.  
  • Pag 8 line 179: “The wavelength scaling in frustrated TIR gives insights into light tunneling regardless of the wavelength.” The authors never say what tunneling is, lease specify and add references
  • Answer. was already discussed in section of frustrated TIR (Fig 1).
  • at pag 11 line 285 (and following) “The high sensitivity ATR detection is realised with multi-reflection slabs/prisms where the number of reflections from the sample-prism interface can increase more than ten times [25].” the sentence here is not clear, please explain
  • Answer. Modified. The increase is due to larger volume probed. It is proportional to number of reflections. 
  • at pag 12: “A toolbox of spectral filters and polarisers are required to fully explore the potential of spectroscopy for material characterisation. Such a toolbox is less developed for the THz spectral range.” less developed compared to what spectral range? can the author explain why?
  • Answer. Compared to visible range. Most probably it is not profitable. 
  • At pag 12: “However, due to low resolution photo-lithography required for longer sub-mm wavelengths, spectral filters can be easily made based on a square lattice” The authors state that the filter can be easly made, do they prepare themself these filters or are them commercially available? This detail should be specified and references should be added (if available).
  • Answer. We made filters ourselves. References for original design are given. Multiplexing of filters is our concept idea and feasibility of fabrication is shown.    
  • “As is known, it is desirable to have a uniform spectral response such that the gain of the bolometer can be precisely set in advance.” The bolometer is mentioned here for the first time, The authors should specify what a bolometer is or at least say before in the text that they used a bolometer as a detector.
  • Answer. Described in sec 2.5.  
  • Pag 13: “cross-filter designs corresponding to different wavenumbers. The first method is a super-pixel type spatial multiplexing where every pixel of the mask is formed by the mosaic of the building block of the different filters. A super-pixel design with 5   5 elements is shown in Fig. 8(a). This is the most straightforward approach but results in undesirable diffraction orders due to the periodic configuration. The second approach uses a random phase mask. In principle, any random phase mask will work but in order to allow at least certain number of crosses of the same type, it is crucial to engineer the scattering ratio of the random phase mask. Such” here there is a total lack of references
  • Answer. There is no lack of references. It is our original proposal.
  • Page 13 line 344: “However, for the unique spectral window at < 3 THz where table top THz sources are not available for high brilliance/intensity,” Thie statement is not precise, there exist Thz laser sources delivering light below 3 Thz.
  • Answer. Edited. 
  • In the Figures caption and in the main text it is often found the plot of “Reflectance vs frequency”. I have two comments about it. First, the authors should show the frequencies always with same units (ether Thz or cm-1), second, is reflectance meant for what is commonly called ATR-absorption? If yes, the authors should clearly explain why they call it reflectance by clearly referring to the experimental setup used.
  • Answer. For analysis of the ATR unit polarisation performance we discuss transmission through all 10 reflections. We have reflections as light passes all ATR setup. It is used in most plain language. We use THz and cm-1 and it is typical in the field.  

Reviewer 2 Report

I would like to thank all the Authors for their efforts.

The authors have answered my comments, for most of the important sections, still the details are missing. The paper is still far away from being a conceptual paper. The authors sum up their previously published works in more like a project report format.  The main idea of creating a hybrid ATR-polarimetry is solely given in the tittle, yet the road map and/or the feasibility and the main know how are not clearly defined and/or explained. For this reason I doubt to recommend publication. On the contrary the reviewed works and the general idea given is interesting and in any case may find use in the scientific community. The work is well organized and described. The introduction provides sufficient background and includes relevant references as a review paper. The concept of the exampled works from the literature are clearly presented and well explained with the supplementary materials.

To this concern following the comments and suggestions given in the previous review report, some major changes are needed. The concept and the scientific know how must be better explained. As a conceptual paper, the concept and the road map to the solution must be better explained. An over all know how on the proposed technique, the possible applications and the potential must be better expressed with relevant references. 

I would like to reconsider after a major revision.

Author Response

I would like to thank all the Authors for their efforts.

The authors have answered my comments, for most of the important sections, still the details are missing. The paper is still far away from being a conceptual paper. The authors sum up their previously published works in more like a project report format.  The main idea of creating a hybrid ATR-polarimetry is solely given in the tittle, yet the road map and/or the feasibility and the main know how are not clearly defined and/or explained. For this reason I doubt to recommend publication. On the contrary the reviewed works and the general idea given is interesting and in any case may find use in the scientific community. The work is well organized and described. The introduction provides sufficient background and includes relevant references as a review paper. The concept of the exampled works from the literature are clearly presented and well explained with the supplementary materials.

To this concern following the comments and suggestions given in the previous review report, some major changes are needed. The concept and the scientific know how must be better explained. As a conceptual paper, the concept and the road map to the solution must be better explained. An over all know how on the proposed technique, the possible applications and the potential must be better expressed with relevant references. 

I would like to reconsider after a major revision.

Answer:  New section on feasibility is added (Sec. 2.5 and fig 7). We carried out polarisation analysis  using synchrotron beam and ATR unit without sample. We prove that the proposed Eqn. 4 can be used to define polarisation response of the ATR unit as hypothesised in the proposed concept.

Other changes were made to express better the main idea of this concept paper. 

We hope that the presented experimental prove of applicability of Eqn. 4 will clarify our aim. We plan to carry out experiments with our samples in the next available beamtime. Beamtimes are granted via a competitive proposal selection.   

Round 3

Reviewer 1 Report

I thank the authors for all the efforts made in improving the quality of the draft. I found the revised version improved and I would recommend it for pubblication after few minor changes.

- page 2 line 45: "Charge separation and its control over the photosynthetic reaction centers upon  illumination and is another active area of research where THz spectroscopy of bio-molecules and
 proteins can provide new insights [4]." Here this sentence is not appropriate. I apologize if my comment in the previous review was not clear, what I meant by suggesting to add references in the draft was to streghten and address the fact that water absorption in THz range strictly depends on temperature; there are wide literature data on this and it is appropriate, for a better comprehension of the topic, to mention it. I suggest the authors also to refer to Andrea MArkelz and MArtina Avenith pubblications on the topic. Tipically, when studying biological samples in THz range, hydration water is one of the component on which researchers focus. This is because the high density of states of biomolecules, i.e. proteins, fatty acids, often hinder the possiility to find narrow structure-related features. In the work of Ludholm et al., the experiments were performed by doing a fine setting of temperature. This is exactly because water absorption changes in a relevant way in temperature and this change could overcross and hide other relevant effects due for example to change in biocomponents structure. Thus, with my previous comment, I wanted to suggest the authors to include the relevant experimental evidences on the topic since water absorption in THz range appears to me to be extremely relevant in the submitted draft.

- In the text is not always clear if the presented topics come from literature or from the authors research work, this is more evident in the section “2.6. Combined spectral filters and polarisers” suggests that it is a review from previous literature data while, as specified by the authors, it is an original concept here presented. I would emphasize this aspect in a more clear way.

Author Response

page 2 line 45: "Charge separation and its control over the photosynthetic reaction centers upon  illumination and is another active area of research where THz spectroscopy of bio-molecules and
 proteins can provide new insights [4]." Here this sentence is not appropriate. I apologize if my comment in the previous review was not clear, what I meant by suggesting to add references in the draft was to streghten and address the fact that water absorption in THz range strictly depends on temperature; there are wide literature data on this and it is appropriate, for a better comprehension of the topic, to mention it. I suggest the authors also to refer to Andrea MArkelz and MArtina Avenith pubblications on the topic. Tipically, when studying biological samples in THz range, hydration water is one of the component on which researchers focus. This is because the high density of states of biomolecules, i.e. proteins, fatty acids, often hinder the possiility to find narrow structure-related features. In the work of Ludholm et al., the experiments were performed by doing a fine setting of temperature. This is exactly because water absorption changes in a relevant way in temperature and this change could overcross and hide other relevant effects due for example to change in biocomponents structure. Thus, with my previous comment, I wanted to suggest the authors to include the relevant experimental evidences on the topic since water absorption in THz range appears to me to be extremely relevant in the submitted draft.

Answer. Discussion extended. 

  • In the text is not always clear if the presented topics come from literature or from the authors research work, this is more evident in the section “2.6. Combined spectral filters and polarisers” suggests that it is a review from previous literature data while, as specified by the authors, it is an original concept here presented. I would emphasize this aspect in a more clear way.

Answer. Clarification is added.

Reviewer 2 Report

I would like to thank all the Authors for their efforts. The authors have answered my comments satisfactorily. I would recommend publication in its recent  form.

With my best regards,

Author Response

Thank you